# Conformational and dynamic plasticity in substrate-binding proteins underlies selective transport in ABC importers

Marijn de Boer[1], Giorgos Gouridis[1,2,3], Ruslan Vietrov[4,5], Stephanie L Begg[6], Gea K Schuurman-Wolters[4,5], Florence Husada[1], Nikolaos Eleftheriadis[1], Bert Poolman[4,5]*, Christopher A McDevitt[6,7]*, Thorben Cordes[1,2]*

[1]Molecular Microscopy Research Group, Zernike Institute for Advanced Materials, University of Groningen, Groningen, The Netherlands; [2]Physical and Synthetic Biology, Faculty of Biology, Ludwig-Maximilians-Universität München, Planegg-Martinsried, Germany; [3]Laboratory of Molecular Bacteriology, Department of Microbiology and Immunology, Rega Institute for Medical Research, KU Leuven, Leuven, Belgium; [4]Department of Biochemistry, Groningen Biomolecular Science and Biotechnology Institute, University of Groningen, Groningen, The Netherlands; [5]Zernike Institute for Advanced Materials, University of Groningen, Groningen, The Netherlands; [6]Department of Microbiology and Immunology, The Peter Doherty Institute for Infection and Immunity, University of Melbourne, Melbourne, Australia; [7]Research Centre for Infectious Diseases, School of Biological Sciences, The University of Adelaide, Adelaide, Australia

*For correspondence:
b.poolman@rug.nl (BP);
christopher.mcdevitt@unimelb.edu.au (CAMD);
cordes@bio.lmu.de (TC)

**Competing interests:** The authors declare that no competing interests exist.

**Abstract** Substrate-binding proteins (SBPs) are associated with ATP-binding cassette importers and switch from an open to a closed conformation upon substrate binding, providing specificity for transport. We investigated the effect of substrates on the conformational dynamics of six SBPs and the impact on transport. Using single-molecule FRET, we reveal an unrecognized diversity of plasticity in SBPs. We show that a unique closed SBP conformation does not exist for transported substrates. Instead, SBPs sample a range of conformations that activate transport. Certain non-transported ligands leave the structure largely unaltered or trigger a conformation distinct from that of transported substrates. Intriguingly, in some cases, similar SBP conformations are formed by both transported and non-transported ligands. In this case, the inability for transport arises from slow opening of the SBP or the selectivity provided by the translocator. Our results reveal the complex interplay between ligand-SBP interactions, SBP conformational dynamics and substrate transport.

DOI: https://doi.org/10.7554/eLife.44652.001

## Introduction

ATP-binding cassette (ABC) transporters facilitate the unidirectional trans-bilayer movement of a diverse array of molecules using the energy released from ATP hydrolysis (*Higgins, 1992*). They share a common architecture, with the translocator unit comprising two transmembrane domains (TMDs) that form the translocation pathway and two cytoplasmic nucleotide-binding domains (NBDs) which bind and hydrolyze ATP. ABC importers require an additional extra-cytoplasmic accessory protein referred to as a substrate-binding protein (SBP) or domain (SBD; hereafter SBDs and SBPs are both termed SBPs) (*Berntsson et al., 2010*; *Scheepers et al., 2016*; *van der Heide and Poolman, 2002*). ABC importers that employ SBPs can be subdivided as Type I or Type II based on

structural and mechanistic distinctions (*Locher, 2016*; *Swier et al., 2016*). A unifying feature of the transport mechanism of Type I and Type II ABC importers is the binding and delivery of substrate from a dedicated SBP to the translocator unit for import into the cytoplasm.

Bacterial genomes encode multiple distinct ABC importers to facilitate the acquisition of essential nutrients such as sugars, amino acids, vitamins, compatible solutes, and metal ions (*Higgins, 1992*; *Davidson et al., 2008*). Many ABC importers can transport more than one type of substrate molecule using high-affinity interactions between SBPs and transported ligands (herein termed cognate substrates) (*Berntsson et al., 2010*). Despite low-sequence similarity between SBPs of different ABC importers, they share a common architecture comprising two structurally conserved rigid lobes connected by a flexible hinge region (*Figure 1*) (*Berntsson et al., 2010*). Numerous biophysical (*Shilton et al., 1996*) and structural analyses (*Quiocho and Ledvina, 1996*) indicate that ligand binding at the interface of the two lobes facilitates switching between two conformations, that is from an open to a closed conformation. Bending and unbending of the hinge region brings the two lobes together (closed conformation) or apart (open conformation), respectively. Crystallographic analyses show that the amount of opening varies between different SBPs; the lobe-movements observed range from small rearrangements as in the Type II SBP BtuF (*Karpowich et al., 2003*), to complete reorientation of both lobes by angles as large as 60° in the Type I SBP LivJ (*Trakhanov et al., 2005*). Nevertheless, the wealth of structural data permits a structural classification of SBPs, wherein the hinge region is the most defining feature of each sub-group or cluster (*Figure 1*) (*Berntsson et al., 2010*; *Scheepers et al., 2016*). Crystal structures of the same protein, but with different ligands bound, generally report the same degree of closing of the SBP (*Trakhanov et al., 2005*; *Nishitani et al., 2012*; *Pandey et al., 2016*; *Magnusson et al., 2004*; *Quiocho et al., 1997*).

Thus, it is assumed that the conformational switching of the SBPs enables the ABC transporter to allosterically sense the loading state of the SBP-ligand complex ('translocation competency'), thereby contributing to transport specificity (*Davidson et al., 2008*; *Quiocho and Ledvina, 1996*).

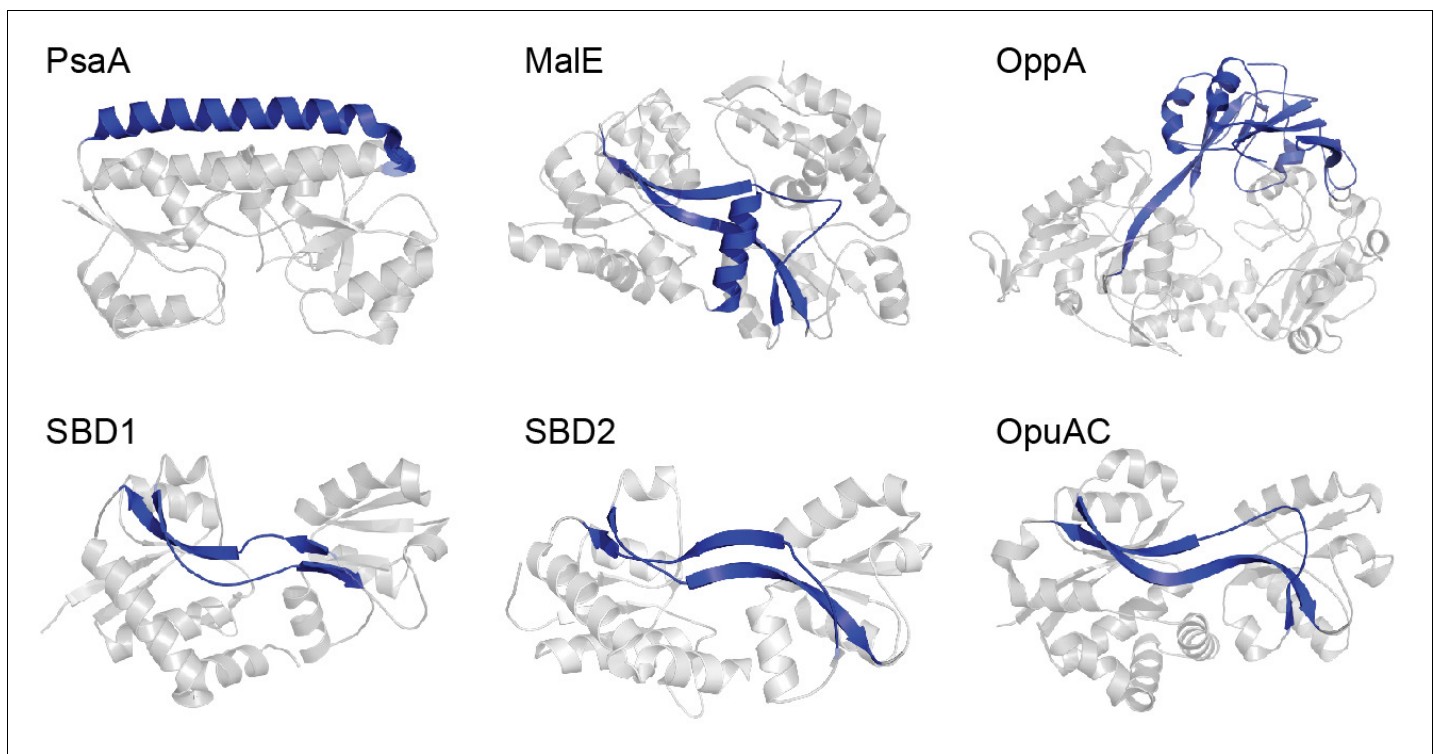

**Figure 1.** Representative SBPs from different structural clusters, categorized by their hinge region. X-ray crystal structures of PsaA (3ZK7; cluster A), MalE (1OMP; cluster B), OppA (3FTO; cluster C), OpuAC (3L6G; cluster F), SBD1 (4LA9; cluster F) and SBD2 (4KR5; cluster F) are all shown in the open, ligand-free conformation. Hinge regions are shown in blue and the two rigid lobes in grey. For classification of the proteins in clusters see (*Berntsson et al., 2010*; *Scheepers et al., 2016*).
DOI: https://doi.org/10.7554/eLife.44652.002

For example, crystal structures of the SBP MalE show that the protein adopts a unique closed conformation when interacting with cognate ligands maltose, maltotriose and maltotetraose (*Quiocho et al., 1997*), while the non-transported ligand β-cyclodextrin is bound by MalE (*Hall et al., 1997a*) but fails to trigger formation of the closed conformation (*Hall et al., 1997b*; *Sharff et al., 1993*; *Skrynnikov et al., 2000*). Ligands that are bound by the SBP, but not transported, are termed herein non-cognate ligands. Such findings suggest that only SBPs which adopt the closed conformation can productively interact with the translocator and initiate transport. However, the TMDs of certain ABC importers were also shown to interact directly with their substrates. In MalFGK$_2$E (*Oldham et al., 2013*) from *Escherichia coli* and Art(QM)$_2$ (*Yu et al., 2015*) from *Thermoanaerobacter tengcongensi* substrate-binding pockets have been identified inside the TMDs, and these might be linked to regulation of transport. Similar binding pockets within the TMDs have not been observed in the high-resolution structures of other ABC importers, although cavities through which the substrate passes in the transition of the TMD from outward- to inward-facing are likely to be present in all the transporters (*Woo et al., 2012*; *Pinkett et al., 2007*; *Locher et al., 2002*). Additional complexity exists for the coupling of SBP conformational switching and the ligand recognition process, as crystallographic (*Flocco and Mowbray, 1994*; *Oswald et al., 2008*), nuclear magnetic resonance (NMR) (*Tang et al., 2007*) and single-molecule (*Feng et al., 2016*; *Gouridis et al., 2015*) studies indicate that SBPs can undergo intrinsic conformational changes in the absence of substrate. Furthermore, crystal structures of the SBPs MalE and a D-xylose SBP were obtained in an open ligand-bound conformation (*Duan and Quiocho, 2002*; *Sooriyaarachchi et al., 2010*). Such observations question the precise relationship between SBP-ligand interactions, SBP conformational changes and their involvement in transport function.

A range of biophysical and structural approaches have been used to decipher the mechanistic basis of SBP-ligand interactions (*Shilton et al., 1996*; *Quiocho and Ledvina, 1996*; *Trakhanov et al., 2005*; *Hall et al., 1997b*; *Skrynnikov et al., 2000*). However, these techniques only provide information on the overall population of molecules. Recent advances in single-molecule methodologies now permit new insight into the conformational heterogeneity, dynamics and occurrences of rare events in SBPs (*Feng et al., 2016*; *Gouridis et al., 2015*; *Kim et al., 2013*; *Seo et al., 2014*; *Husada et al., 2015*; *Lerner et al., 2018*), which are difficult to obtain in bulk measurements. Here, we combined single-molecule Förster resonance energy transfer (smFRET) (*Ha et al., 1996*) and transport measurements to investigate how cognate and non-cognate substrates influence the conformational states and the underlying dynamics of SBPs. Six distinct SBPs were selected (*Figure 1*) (*Fulyani et al., 2016*; *Wolters et al., 2010*; *Ferenci, 1980*; *McDevitt et al., 2011*; *Berntsson et al., 2011*), based on two criteria. First, they cover the breadth of SBP structural classes: PsaA (cluster A), MalE (cluster B), OppA (cluster C), SBD1 and SBD2 of GlnPQ, and OpuAC (all cluster F). The selected SBPs provide coverage of hinge region diversity (*Berntsson et al., 2010*; *Scheepers et al., 2016*), thereby addressing a hypothesized key determinant in SBP conformational dynamics. Moreover, subtle structural or sequence differences among SBPs that belong to the same cluster are addressed by examining SBD1, SBD2 and OpuAC that all belong to cluster F. Second, the selected SBPs belong to Type I and Type II ABC importers with extensively characterized substrate (cognate and non-cognate) interactions, such as metal ions (PsaA) (*McDevitt et al., 2011*), sugars (MalE) (*Ferenci et al., 1986*), peptides (OppA) (*Doeven et al., 2004*), amino acids (SBD1 and SBD2) (*Fulyani et al., 2016*), and compatible solutes (OpuAC) (*Wolters et al., 2010*).

## Results

### Multiple SBP conformations are translocation competent

Crystal structures of SBPs suggest that ligand binding is coupled to switching between an open and a closed conformation. Mechanistically, this process has been linked to the allosteric regulation of substrate transport (*Davidson et al., 2008*; *Shilton et al., 1996*; *Quiocho and Ledvina, 1996*; *Oldham and Chen, 2011*; *Hor and Shuman, 1993*; *Doeven et al., 2008*; *Hollenstein et al., 2007*; *Davidson et al., 1992*). Here, we assessed this model by investigating the interaction of six SBPs, PsaA, MalE, OppA, SBD1, SBD2 and OpuAC, with a range of cognate substrates. We employed single-molecule FRET to analyze SBP conformations, wherein each of the two SBP lobes was labeled with either a donor or an acceptor fluorophore (*Figure 2A*) (*Gouridis et al., 2015*; *Kapanidis et al.,*

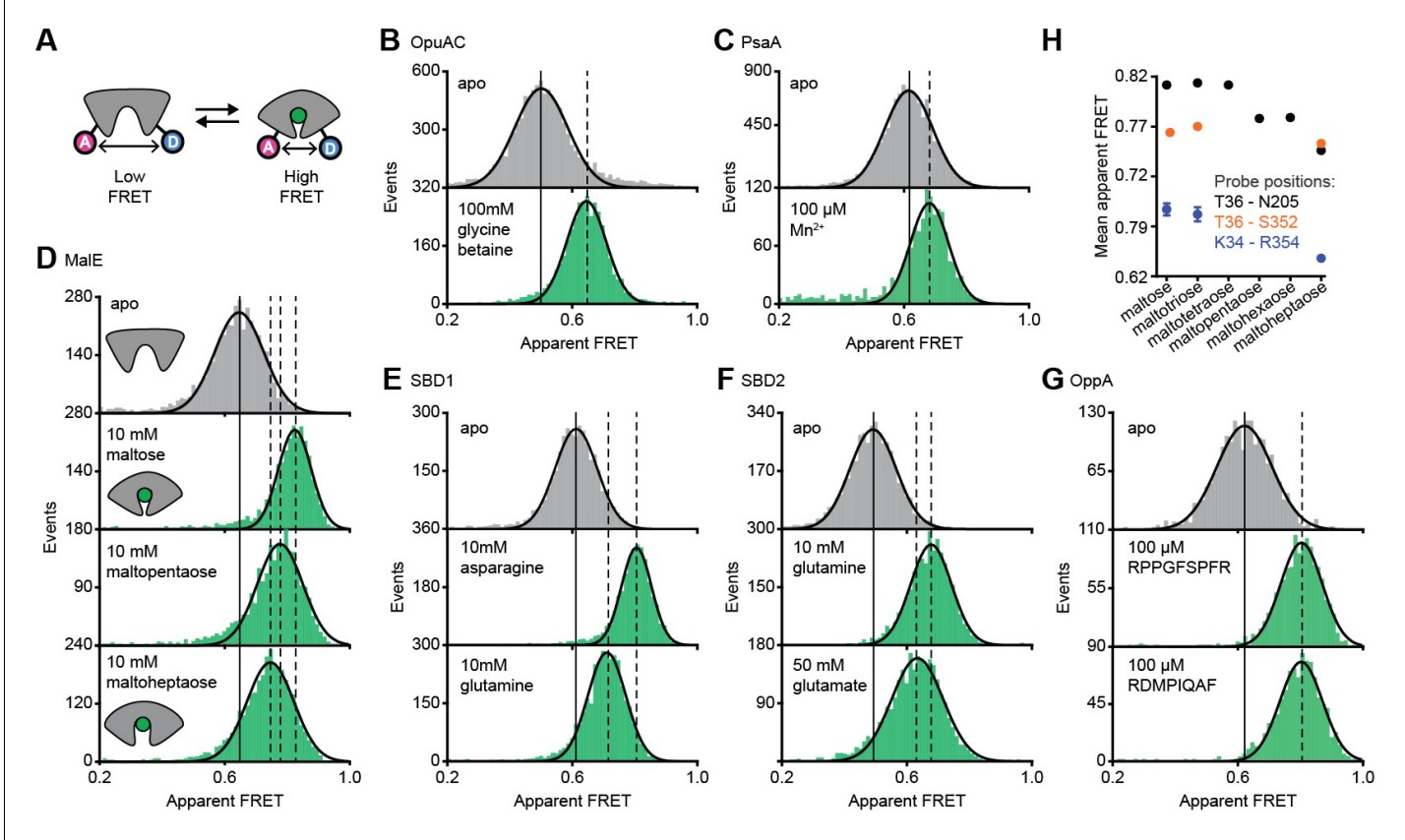

**Figure 2.** Conformational states of SBPs probed by smFRET reveal multiple active conformations. (A) Experimental strategy to study SBP conformational changes via FRET. Solution-based apparent FRET efficiency histograms of OpuAC(V360C/N423C) (B), PsaA(V76C/K237C) (C), MalE (T36C/S352C) (D), SBD1(T159C/G87C) (E), SBD2(T369C/S451) (F) and OppA(A209C/S441C) (G) in the absence (grey bars) and presence of different cognate substrates (green bars). The OppA substrates are indicated by one-letter amino acid code. Bars are the data and the solid line a Gaussian fit. The 95% confidence interval of the Gaussian distribution mean is shown in *Supplementary file 3*, and the interval center is indicated by vertical lines (solid and dashed). (H) Mean of the Gaussian distribution of MalE labeled at T36/S352 (black), T36/N205 (green) or K34/R352 (blue). Error bars indicate 95% confidence interval.

DOI: https://doi.org/10.7554/eLife.44652.005

The following source data and figure supplements are available for figure 2:

**Source data 1.** Apparent FRET efficiency histograms of *Figure 2B–G*.

DOI: https://doi.org/10.7554/eLife.44652.010

**Source data 2.** Apparent FRET efficiency histograms of *Figure 2—figure supplement 3*.

DOI: https://doi.org/10.7554/eLife.44652.011

**Figure supplement 1.** Ligand-induced conformational dynamics of SBPs.

DOI: https://doi.org/10.7554/eLife.44652.006

**Figure supplement 2.** OppA uses an induced-fit ligand binding mechanism.

DOI: https://doi.org/10.7554/eLife.44652.007

**Figure supplement 3.** Translocation competent conformation(s) of MalE and OppA.

DOI: https://doi.org/10.7554/eLife.44652.008

**Figure supplement 4.** MalE conformations studied by smFRET.

DOI: https://doi.org/10.7554/eLife.44652.009

*2004*). Surface-exposed and non-conserved residues, showing largest distance changes according to the crystal structures of the open and closed states, were selected as suitable cysteine positions for labeling. Labeling and surface-immobilization of the protein molecules did not alter the ligand dissociation constant $K_D$ (*Table 1*). In our assays, the inter-dye distance reports on the relative orientation and distance between the SBP lobes and is thus indicative for the degree of closing. Steady-state anisotropy measurements indicate that the dyes retain sufficient rotational freedom (*Table 2*) so that

**Table 1.** Dissociation constant $K_D$ of substrate-binding proteins.

| Protein[*] | Ligand | $K_D$ (μM) Freely-diffusing protein | Surface-tethered protein | $K_D$ WT protein[¶] (μM) |
|---|---|---|---|---|
| OpuAC(V360C/N423C) | Glycine betaine | $3.4 \pm 0.4$[†] | $3.1$[‡] | 4–5 (*Wolters et al., 2010*) |
| OppA(A209C/S441C) | RPPGFSFR | $7.0 \pm 1$[†] | $14 \pm 5$[#] | $5 \pm 3$[#] |
| SBD2(T369C/S451) | Glutamine | $1.2 \pm 0.2$[§] | $0.5$[‡] | $0.9 \pm 0.1$ (*Gouridis et al., 2015*) |
| SBD1(T159C/G87C) | Asparagine | $0.34 \pm 0.03$[§] | $0.3$[‡] | $0.2 \pm 0.0$ (*Gouridis et al., 2015*) |
| MalE(T36C/S352C) | Maltose | $1.7 \pm 0.3$[†] | $2.2$[‡] | 1-2 (*Hall et al., 1997a, Kim et al., 2013*) |
| MalE(T36C/S352C) | Maltotriose | $0.6 \pm 0.2$[†] | $0.9$[‡] | 0.2-2 (*Hall et al., 1997a, Kim et al., 2013*) |

[*]. $K_D$ could not be determined reliably for labeled PsaA due to background metal contamination.

[†]. Population of the closed conformation $P$ in the presence of a ligand concentration $L$ was determined using solution-based smFRET. The $K_D = L\,(1 - P)/P$ for a one-binding site model. Data corresponds to mean ± s.d. of duplicate experiments with the same protein sample.

[‡]. *Figure 2—figure supplement 1*

[§]. *Figure 4—figure supplement 2*

[#]. *Figure 2—figure supplement 2*

[¶]. The $K_D$ values of wildtype (WT) proteins are obtained from the indicated references.

DOI: https://doi.org/10.7554/eLife.44652.003

relative inter-dye distance can be assessed via the apparent FRET efficiency of freely diffusing or surface-immobilized protein molecules. Although this approach monitors only a single distance in the SBP, it permits rapid screening of ligand induced conformational changes under physiologically relevant conditions.

The apparent FRET efficiency distributions of individual and freely diffusing SBPs were determined in the presence and absence of their cognate substrates using confocal microscopy. Saturating concentrations of cognate substrate, above the $K_D$ (*Table 1*), shift the FRET efficiency histograms and the fitted Gaussian distributions to higher values compared to the ligand-free SBPs (*Figure 2B–G*; *Supplementary file 3*), indicating a reduced distance between the SBP lobes and inferred to be closure of the proteins. For individual surface-immobilized SBPs, we observed ligand-induced opening and closing transitions in the presence of ligand concentrations at the respective $K_D$ value (*Figure 2—figure supplement 1*). The solution-based FRET distributions of ligand-bound and ligand-free SBPs are unimodal and thus do not reveal any substantial conformational heterogeneity, such as a pronounced closing in the absence of substrate or a substantial population of an open-liganded state (*vide infra*). This strongly suggests that ligands are bound via an induced-fit mechanism, unless dynamics occur on timescales faster than milliseconds. This inference was confirmed for OppA by examining individual surface-immobilized proteins and demonstrating that

**Table 2.** Steady-state anisotropy values.

| | Anisotropy Alexa555 | Alexa647 | Cy3B | Atto647N |
|---|---|---|---|---|
| Free dye | 0.25 | 0.20 | 0.08 | 0.08 |
| OpuAC(V360C/N423C) | NA | NA | 0.17 | 0.11 |
| OppA(A209C/S441C) | 0.25 | 0.19 | NA | NA |
| SBD1(G87C/T159C) | 0.27 | 0.19 | NA | NA |
| SBD2(T369C/S451) | 0.26 | 0.20 | NA | NA |
| MalE(T36C/S352C) | 0.29 | 0.24 | NA | NA |
| PsaA(V76C/K237C) | 0.28 | 0.22 | NA | NA |

NA: not applicable. Data correspond to mean (s.d. below < 0.01) of duplicate experiments, using the same labeled protein sample.

DOI: https://doi.org/10.7554/eLife.44652.004

substrate-induced SBP closing follows first-order kinetics while the opening obeys zeroth-order kinetics (*Figure 2—figure supplement 2*) (*Kim et al., 2013*).

Further examination of the FRET distributions shows that multiple substrate-bound SBP conformations exist for SBD1, SBD2 and MalE (*Figure 2D–F*). For the amino acid binding-proteins SBD1 and SBD2, the cognate substrates (*Fulyani et al., 2016*) asparagine and glutamine for SBD1, and glutamine and glutamate for SBD2 all stabilize a distinct protein conformation, as shown by the FRET efficiency histograms and fitted Gaussian distributions (*Figure 2E–F*; *Supplementary file 3*). Notably, closure of SBD1 by asparagine reduces the inter-dye distance compared to the ligand-free protein by ~9 Å (*Supplementary file 3*). In contrast, glutamine binding reduces the distance by ~5 Å, suggesting that only a partial closing of SBD1 occurs. In SBD2, glutamine and glutamate reduce the distance ~9 and ~7 Å, respectively (*Supplementary file 3*).

For the maltodextrin binding-protein MalE, we examined the effect of cognate maltodextrins (*Ferenci, 1980*), ranging from two to seven glucosyl units, on the protein conformation. Comparison of the FRET efficiency histograms of the different MalE-ligand complexes shows that at least three distinct ligand-bound MalE conformations exist (*Figure 2D*; *Figure 2—figure supplement 3A*; *Supplementary file 3*). In contrast to SBD1 and SBD2, some cognate substrates did not induce a unique MalE conformation (*Figure 2—figure supplement 3A*). For example, maltopentaose and maltohexaose elicited the same FRET change, and triggered the formation of a partially closed MalE conformation with a ~7 Å reduction in the inter-dye distance. This conformational state is different from the fully closed form of MalE, induced by maltose, maltotriose and maltotetraose, wherein the inter-dye distance is reduced by ~10 Å. Further, it is also distinct from the other partially closed conformation induced by maltoheptaose where the inter-dye distance is reduced by ~5 Å. These results were confirmed by examining different inter-dye distances (*Figure 2H*; *Figure 2—figure supplement 4*). However, whether this conformational plasticity is a universal feature among SBPs needs to be investigated further, because in OppA the four examined cognate substrates (*Doeven et al., 2004*) elicited the same FRET change (*Figure 2G*; *Figure 2—figure supplement 3B*). The findings on the conformational changes (and differences) for each SBP were shown to be statistically robust by the non-parametric two-way Kolmogorov-Smirnov (KS) test (p-values in *Supplementary file 1*), which indicates the absence of any fitting bias. Taken together, these data indicate that although the examined SBPs have a single open conformation, a productive interaction between the SBP and the translocator does not require a single, unique closed SBP conformation. The structural flexibility of the SBP permits the formation of one or more ligand-bound conformations, all of which are able to interact with the translocator and initiate transport (*Fulyani et al., 2016*; *Wolters et al., 2010*; *Ferenci, 1980*; *McDevitt et al., 2011*; *Doeven et al., 2004*).

## Intrinsic conformational changes of SBPs

We then investigated whether the conformational changes in the SBPs that were triggered by their ligands, can also occur in their absence. To address this, we investigated surface-tethered SBPs in the absence of ligand and used confocal scanning microscopy to obtain millisecond temporal resolution. Compared to the solution-based smFRET experiments, individual surface-tethered SBPs greatly increase the sensitivity to detect rare events. In contrast to prior work (*Feng et al., 2016*; *Gouridis et al., 2015*; *Kim et al., 2013*; *Seo et al., 2014*), the labeled SBPs were supplemented with high concentrations of unlabeled protein (10–20 μM), or the divalent chelating compound ethylenediaminetetraacetic acid (1 mM EDTA for PsaA), to remove any contaminating ligands (*Figure 3A*). Contaminations could otherwise lead to conformational changes that are misinterpreted as intrinsic closing of the SBP. Consistent with the solution-based measurements, all SBPs were predominantly in a low FRET state (open conformation; *Figure 3B–G*; *Figure 3—figure supplement 1*). For ligand-free MalE, PsaA and OpuAC, no transitions to higher FRET states were observed within a total observation time of >8 min for each SBP (*Figure 3B–D*; *Supplementary file 4*). In SBD1, SBD2 and OppA rare transitions to a high FRET state can be observed and have an average lifetime of 205 ± 36, 90 ± 11 and 211 ± 42 ms (mean ± s.e.m.), respectively (*Figure 3E–G*; *Figure 3—figure supplement 1D–F*). Transitions toward these states occur only rarely, that is, on average 2–8 times per minute (*Figure 3H*; *Supplementary file 4*). To rule out that these infrequent FRET transitions are caused by rare binding events arising from any non-chelated ligand, we analyzed the protein conformational dynamics of SBD1, SBD2 and OppA in the presence of a 4 to 10-fold lower concentration of unlabeled protein. We observed that the FRET transitions occur with a

similar frequency and have the same average lifetime compared to when 10–20 µM unlabeled protein is present (*Figure 3—figure supplement 2*). This suggests that all potential ligand contamination is efficaciously scavenged by unlabeled protein, thus providing compelling evidence that the rare FRET transitions observed in SBD1, SBD2 and OppA represent intrinsic closing of the protein. Therefore, some SBPs have the ability to also close without the ligand on the second timescale. However, not all SBPs show intrinsic conformational transitions, unless these occur below the temporal resolution of the measurements (millisecond timescale). Overall, the data indicate that diversity exists in the conformational dynamics of ligand-free SBPs.

## How do non-transported substrates influence the SBP conformation?

Ensemble FRET measurements using all proteinogenic amino acids and citruline were performed to obtain full insight into substrate specificity of SBD1 and SBD2 of GlnPQ. We find that asparagine, glutamine and histidine elicit a FRET change in SBD1, and glutamine in SBD2 (*Figure 4—figure supplement 1*); glutamate triggers a change in SBD2 at low pH, that is, when a substantial fraction of glutamic acid is present. No other amino acid affected the apparent FRET efficiency. However, arginine and lysine competitively inhibit the conformational changes induced by asparagine binding to SBD1 and glutamine binding to SBD2 (*Figure 4—figure supplement 2*). Uptake experiments in whole cells and in proteoliposomes show that histidine, lysine and arginine are not transported by GlnPQ, but these amino acids can inhibit the uptake of glutamine (via SBD1 and SBD2) and asparagine (via SBD1) (*Figure 4A–C*). Thus, some amino acids interact with the SBPs of GlnPQ but fail to trigger transport. Similar ligands have been identified for MalE, OpuAC and PsaA (*Hall et al., 1997a*; *Wolters et al., 2010*; *Ferenci, 1980*; *McDevitt et al., 2011*), and we refer to these as non-cognate substrates. We then used smFRET to test whether or not ligand-induced SBP conformational changes allow discrimination of cognate from non-cognate substrates.

At saturating concentrations of most non-cognate ligands the FRET efficiencies are altered compared to the ligand-free conditions (*Figure 4D–H*; *Supplementary file 3*; *Supplementary file 1*). This shows that, similar to cognate ligands (*Figure 3B–G*), non-cognate ligand binding is coupled to SBP conformational changes. However, this is not true in all cases, as the binding of the non-cognate substrates, that is, arginine or lysine for SBD1 and arginine for SBD2 do not alter the FRET efficiency histograms (*Figure 4D–E*), suggesting that these ligands bind in the open conformation of the SBP and do not trigger a conformational change.

Further analysis of the non-cognate ligand-induced conformational changes reveals states that vary, from a minor opening (carnitine-OpuAC in *Figure 4G*), to partial (histidine-SBD1 in *Figure 4D*; various maltodextrin-MalE complexes in *Figure 4F*; proline-OpuAC in *Figure 4G*) or full closing ($Zn^{2+}$-PsaA in *Figure 4H*) of the SBP relative to the ligand-free state of the corresponding protein. The data of full closing by $Zn^{2+}$ (non-cognate) and $Mn^{2+}$ (cognate) were confirmed by examining different inter-dye positions in PsaA (*Figure 4—figure supplement 3*) and are in line with prior crystallographic analyses (*McDevitt et al., 2011*; *Lawrence et al., 1998*). Noteworthy, the non-cognate substrate histidine and the cognate substrate glutamine induce both partial closing of SBD1 (*Figure 4D*). However, histidine elicited a larger FRET shift in SBD1 (~7 Å reduction in inter-dye distance) than cognate glutamine (~5 Å), but smaller than the cognate substrate asparagine (~9 Å), which induced full closing (*Figure 4D*, *Supplementary file 3*). In contrast, the FRET shift induced with certain non-cognate ligands in MalE (β-cyclodextrin, maltotriitol and maltotetraitol) and OpuAC (proline) are smaller (or similar; *vide infra*) than with their cognate ligands (*Figure 4F–G*), which corresponds with a reduction in the inter-dye distance of ~3–4 Å, in contrast to ~9–10 Å for full closure of these SBPs (*Supplementary file 3*). Intriguingly, the data also suggest that the partially closed SBP-ligand complexes of MalE formed with the non-cognate substrates maltooctaose or maltodecaose are similar to that of the cognate substrate maltoheptaose (*Figure 4F*). Again, this result was confirmed by examining different inter-dye positions in MalE (*Supplementary file 3*). The findings on the conformational changes (and differences) for each SBP were shown to be statistically robust by the two-way KS test (*Supplementary file 1*).

In summary, similar to cognate substrates, non-cognate substrates do not induce a single unique ligand-bound SBP state, and solely from the degree of SBP closing a translocator cannot readily discriminate cognate from non-cognates substrates. Notable exceptions are the substrates that do not induce closing and keep the SBP in the open state. This raises fundamental questions as to the mechanistic basis for how certain non-cognate substrates are still excluded from import.

## Altered SBP opening renders PsaA permissive for non-cognate ligand transport

The inability of certain substrates to be transported, while they appear to induce SBP conformations that are similar to those associated with cognate substrates, was observed for MalE (*Figure 4F*) and PsaA (*Figure 4H*). First, this was investigated further for PsaA. Upon addition of 1 mM EDTA to PsaA-$Mn^{2+}$, lower FRET efficiencies are instantaneously recorded (*Figure 5A*), indicating that the lifetime of the closed PsaA-$Mn^{2+}$ conformation is shorter than a few seconds. By contrast, $Zn^{2+}$ kept PsaA closed, irrespective of the duration of the EDTA treatment (up to 15 min) (*Figure 5B*). Irreversible and reversible binding of these metals was shown previously (*Couñago et al., 2014*), which can now be explained by the fast and slow opening of PsaA in the presence of $Mn^{2+}$ and $Zn^{2+}$, respectively. The extremely slow opening of PsaA may explain why $Zn^{2+}$ is not transported by PsaBCA, as opening of the SBP is required for release of the ligand to the translocator. However, it is also possible that the translocator controls the transport specificity (*Oldham et al., 2013*; *Yu et al., 2015*). To discriminate between these two scenarios, we examined the impact of altered SBP dynamics on the transport activity of PsaBC. We substituted an aspartate in the binding site with asparagine (D280N), which has previously been shown to perturb the stability of the $Zn^{2+}$-bound SBP (*Couñago et al., 2014*). Analysis of PsaA and PsaA(D280N), at saturating $Zn^{2+}$ concentrations, revealed similar FRET efficiency histograms for the two proteins (*Figure 5C*; *Supplementary file 3*). However, in contrast to the $Zn^{2+}$-PsaA complex, opening of the PsaA(D280N) complex renders $Zn^{2+}$ accessible to EDTA, similar to the cognate ligand $Mn^{2+}$ (*Figure 5A,C*). The ability of PsaA(D280N) to open and release $Zn^{2+}$ was then assessed by measuring the cellular accumulation of $Zn^{2+}$ within *Streptococcus pneumoniae*, the host organism. This was achieved by replacement of the *psaA* gene with the D280N mutant allele ($\Omega psaA_{D280N}$) in a strain permissive for $Zn^{2+}$ accumulation, that is incapable of $Zn^{2+}$ efflux due to deletion of the exporter CzcD ($\Omega psaA_{D280N}\Delta czcD$) (*Begg et al., 2015*). Our data show that cellular $Zn^{2+}$ accumulation increases in the strain expressing PsaBC with PsaA(D280N) but not with wild-type PsaA (*Figure 5D*). These results demonstrate that the altered conformational dynamics of the PsaA derivative renders ligand release permissive for transport of non-cognate $Zn^{2+}$ ions. The data also show that translocator activity is not directly influenced by the nature of the metal ion released by PsaA. Collectively, our findings show that transport specificity of PsaBCA is dictated by the opening kinetics of PsaA.

## MalE conformational dynamics with cognate and non-cognate substrates

Next, we determined the conformational dynamics of MalE induced by maltoheptaose, maltooctaose and maltodecaose. Similar to $Zn^{2+}$ and $Mn^{2+}$ in PsaA (*Figure 4H*), these substrates appear to induce similar MalE conformations (*Figure 4F*) but only maltoheptaose is transported (*Ferenci, 1980*). Measurements on individual surface-tethered MalE proteins, in the presence of maltoheptaose, maltooctaose or maltodecaose, show frequent switching between low and higher FRET states, corresponding to opening and (partial) closing of MalE (*Figure 6A–D*). Consistent with the solution-based smFRET measurements, the average apparent FRET efficiency of the high FRET state is similar for these maltodextrins and lower than with maltose (*Figure 6—figure supplement 1*). The mean lifetime of the ligand-bound conformations (mean lifetime of the high FRET states) are $328 \pm 8$ ms for cognate maltoheptaose and $319 \pm 12$ ms and $341 \pm 8$ ms for non-cognate maltooctaose and maltodecaose, respectively (mean $\pm$ s.e.m.; *Figure 6A*, *Figure 6—figure supplement 2*). So, contrary to PsaA-$Zn^{2+}$ (*Figure 5*), a slow opening of MalE and inefficient ligand release kinetics cannot explain why maltooctaose and maltodecaose are not transported; the average lifetimes with maltooctaose or maltodecaose are not significantly different from that with maltoheptaose (p = 0.68, one-way analysis of variance (ANOVA); *Figure 6A*). Most likely, the failure of the maltose system to transport maltooctaose and maltodecaose originates from the size limitations of the translocator domain of MalFGK$_2$ (*Oldham et al., 2013*).

## Translocator/SBP interplay determines the rate of transport

Finally, we sought to elucidate the mechanistic basis for how substrate preference arises in the maltose system and to what degree the translocator contributes to this process. First, we investigated how the MalE conformational dynamics influences the transport rate of the substrate maltose. For

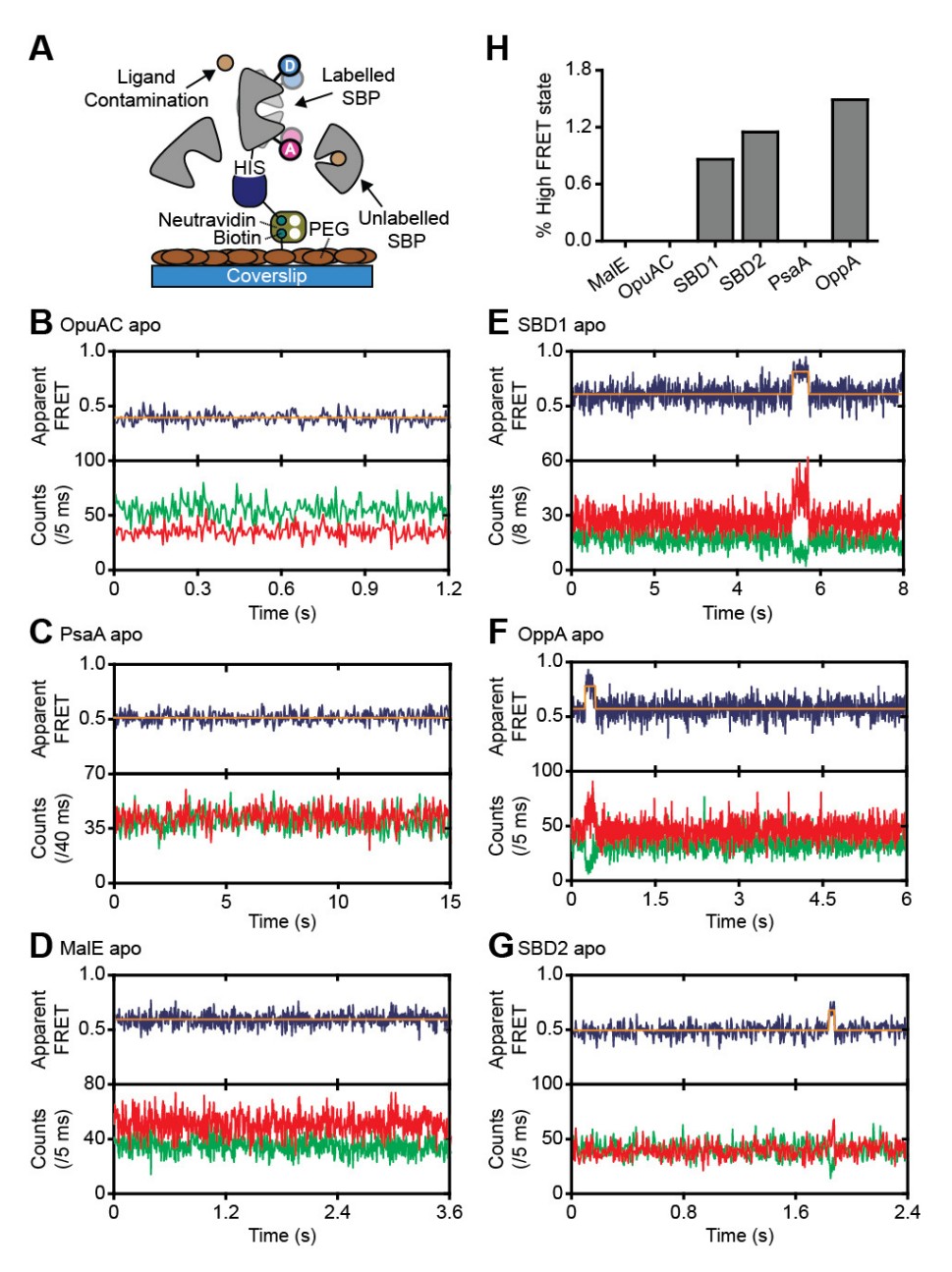

**Figure 3.** Rare conformational states of ligand-free SBPs. (**A**) Schematic of the experimental strategy to study the conformational dynamics of ligand-free SBPs. Representative fluorescence trajectories of OpuAC(V360C/N423C) (**B**), PsaA(V76C/K237C) (**C**), MalE(T36C/S352C) (**D**), SBD1(T159C/G87C) (**E**), OppA(A209C/S441C) (**F**) and SBD2 (T369C/S451) (**G**) in the absence of substrate. 10–20 µM of unlabeled protein or 1 mM EDTA (for PsaA) was added to scavenge any ligand contaminations. In all fluorescence trajectories presented in the figure: top panel shows calculated apparent FRET efficiency (blue) from the donor (green) and acceptor (red) photon counts as shown in the bottom panels. Orange lines indicate average apparent FRET efficiency value or most probable state-trajectory of the Hidden Markov Model (HMM). Statistics in *Supplementary file 4*. (**H**) Percentage of time a SBP is in the high FRET state. Statistics in *Supplementary file 4*.

DOI: https://doi.org/10.7554/eLife.44652.012

The following source data and figure supplements are available for figure 3:

**Source data 1.** Donor and acceptor photon counts, apparent FRET efficiency and most probable state-trajectory of the Hidden Markov Model of the traces in *Figure 3*.

DOI: https://doi.org/10.7554/eLife.44652.015

*Figure 3 continued on next page*

*Figure 3 continued*

**Figure supplement 1.** Conformational dynamics of ligand-free and ligand-bound SBPs.
DOI: https://doi.org/10.7554/eLife.44652.013
**Figure supplement 2.** Intrinsic conformational dynamics in the presence of unlabeled protein.
DOI: https://doi.org/10.7554/eLife.44652.014

this we used the hinge-mutant variant MalE(A96W/I329W) that has different conformational dynamics compared to the wild-type protein (*Figure 6E*; *Figure 6—figure supplement 3A–B*) (*Kim et al., 2013*). The mutations are believed to not affect SBP-translocator interactions since they are situated on the opposite side of the interaction surface of the SBP (*Oldham and Chen, 2011*; *Gould et al., 2009*).

At saturating concentrations of maltose, the FRET efficiency distributions of MalE and MalE (A96W/I329W) are indistinguishable (*Figure 6—figure supplement 3C*). This could be confirmed by two different inter-dye positions in each protein. Therefore, changes in the rate of maltose transport unlikely arise from differences in SBP docking onto the TMD, since similar SBP conformations are involved. Nonetheless, cellular growth and the maltose-induced ATPase activity are reduced for MalE(A96W/I329W) (*Gould et al., 2009*; *Bao and Duong, 2012*). Analysis of the mean lifetime of the closed conformation of MalE(A96W/I329W) shows that the opening of the protein is almost three orders of magnitude slower than in the wild-type protein [63 ± 6 ms (mean ±s.e.m.) in MalE *versus* 28 ± 5 s (mean ± s.e.m.) in MalE(A96W/I329W); *Figure 6A*; *Figure 6—figure supplement 3B*]. These observations suggest that the maltose-stimulated cellular growth and ATPase activity are reduced due to the slower ligand release of MalE(A96W/I329W) compared to wildtype MalE. This negative correlation between the MalE lifetime and the transport activity is in line with the observation that $Zn^{2+}$-PsaA(D280N) opens fast, so that $Zn^{2+}$ transfer to the translocator and import can occur, whereas in wildtype $Zn^{2+}$-PsaA the opening is (extremely) slow and import does not occur (*Figure 5B–D*).

We then investigated the relationship between maltodextrin-specific lifetimes of the MalE closed conformations and published transport rates or ATPase activities of the full transport system (*Hall et al., 1997a*). Here, we focused on the cognate substrates maltose, maltotriose and maltotetraose. Analysis of individual surface-tethered MalE proteins in the presence of these substrates shows that the average lifetime of the closed conformation with maltose, maltotriose and maltotetraose are 63 ± 6, 124 ± 4, and 150 ± 8 ms (mean ± s.e.m.), respectively (*Figure 6A*; *Figure 6E–G*; *Figure 6—figure supplement 2*). Thus, these lifetimes correlate positively with their stimulation of the ATPase activity (*Figure 6H*) (*Hall et al., 1997a*). A positive relationship also exists between the lifetimes with maltose and maltotetraose (63 ± 6 and 150 ± 8 ms, respectively) and their corresponding transport rates (transport of maltotetraose is ~1.5 fold higher than of maltose) (*Hall et al., 1997a*). This positive correlation is inconsistent with our earlier findings that a shorter SBP lifetime results in a faster rate of transport. However, this relationship only holds when the SBP conformational dynamics are altered, while leaving all other rate-determining steps of the transport process unaffected. Thus, the observation that some maltodextrins induce a faster opening of MalE, while their corresponding transport and/or stimulation of ATP hydrolysis are slower, implies that the kinetics of certain other rate-determining steps are substrate-dependent. Faster transport or ATP hydrolysis can arise when certain maltodextrins trigger these steps more efficiently than others, thereby overcoming the slower opening of MalE. These steps most likely occur after opening of MalE, as these differences in transport activity are unlikely to arise from differences in docking of MalE onto the TMDs (crystallographic (*Quiocho et al., 1997*) and smFRET data (*Supplementary file 3*) shows that maltose, maltotriose and maltotetraose induce similar MalE conformations) or the differences in the binding affinity of MalE (*Hall et al., 1997a*). Thus, although the precise molecular mechanism of the rate-determining steps remains elusive, the positive correlation between lifetime of the SBP closed conformation and the activity of the transporter strongly suggests involvement of the translocator MalFGK$_2$ in influencing the transport rate of certain maltodextrins.

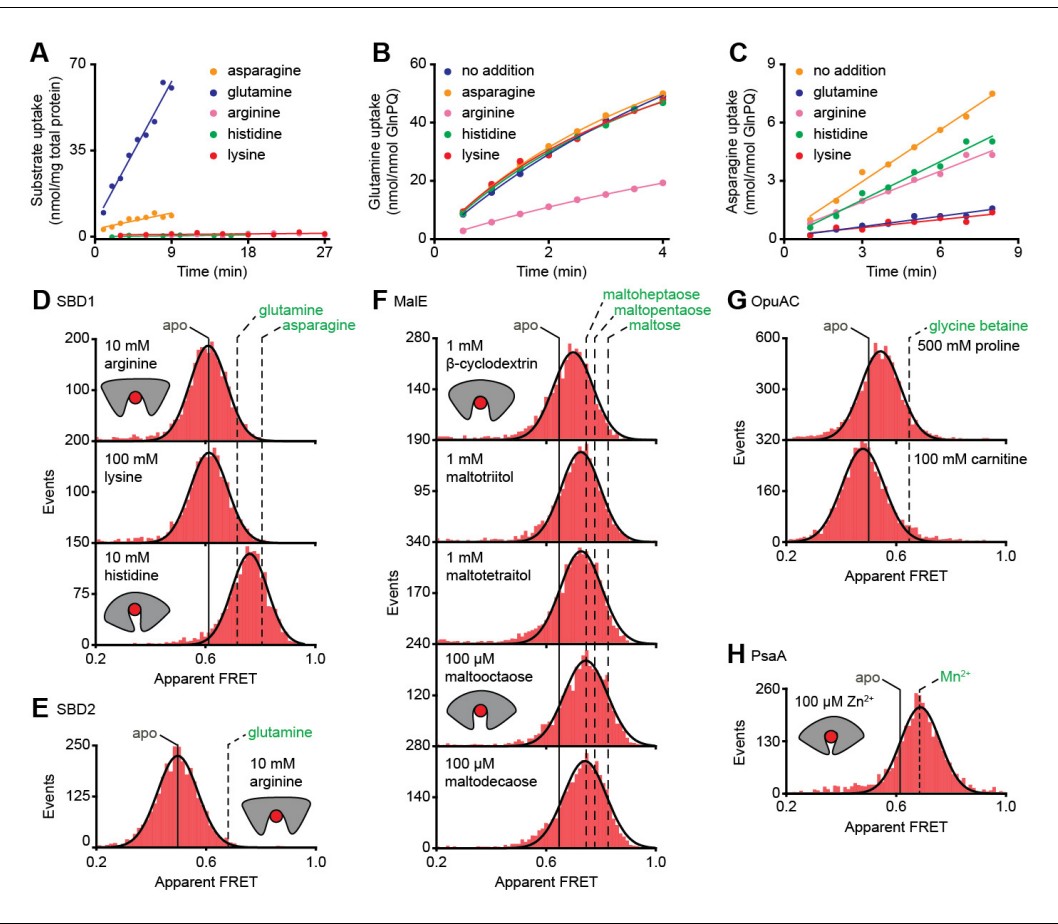

**Figure 4.** Substrate-specificity of GlnPQ and SBP conformations induced by non-cognate substrates. (**A**) Time-dependent uptake [$^{14}$C]-asparagine (5 µM), [$^{14}$C]-glutamine (5 µM), [$^{14}$C]-arginine (100 µM), [$^{14}$C]-histidine (100 µM) and [$^{3}$H]-lysine (100 µM) by GlnPQ in *L. lactis* GKW9000 complemented in *trans* with a plasmid for expressing GlnPQ; the final amino acid concentrations are indicated between brackets. Points are the data and the solid line a hyperbolic fit. Time-dependent uptake of glutamine (**B**) and asparagine (**C**) in proteoliposomes reconstituted with purified GlnPQ (see Materials and methods section). The final concentration of [$^{14}$C]-glutamine and [$^{14}$C]-asparagine was 5 µM, respectively; the amino acids indicated in the panel were added at a concentration of 5 mM. Solution-based apparent FRET efficiency histogram of SBD1(T159C/G87C) (**D**), SBD2(T369C/S451) (**E**), MalE(T36C/S352C) (**F**), OpuAC(V360C/N423C) (**G**) and PsaA(V76C/K237C) (**H**) in the presence of non-cognate (red bars) substrates as indicated. Bars are the data and solid line a Gaussian fit. The 95% confidence interval for the distribution mean is shown in *Supplementary file 3*. The interval center is indicated by vertical lines (solid and dashed).

DOI: https://doi.org/10.7554/eLife.44652.016

The following source data and figure supplements are available for figure 4:

**Source data 1.** Apparent FRET efficiency histograms of *Figure 4D–H*.
DOI: https://doi.org/10.7554/eLife.44652.020
**Figure supplement 1.** Substrate binding of SBD1 and SBD2 studied by ensemble FRET.
DOI: https://doi.org/10.7554/eLife.44652.017
**Figure supplement 2.** Non-cognate substrate binding by SBD1 and SBD2.
DOI: https://doi.org/10.7554/eLife.44652.018
**Figure supplement 3.** PsaA(E74C/K237C) conformational changes probed by smFRET.
DOI: https://doi.org/10.7554/eLife.44652.019

## Discussion

Prokaryotes occupy diverse ecological niches within terrestrial ecosystems. Irrespective of the niche, their viability depends on selective acquisition of nutrients from the extracellular environment.

However, the diversity of the external milieu poses a fundamental challenge for how acquisition of specific compounds can be achieved within the constraints of the chemical selectivity conferred by their import pathways. Numerous studies on SBPs associated with ABC importers have established that these proteins share a common architecture with a well-defined high-affinity ligand-binding site and have the ability to adopt a distinct ligand-free and -bound conformation, that is open and closed, respectively (*Berntsson et al., 2010*; *Davidson et al., 2008*; *Shilton et al., 1996*). Building on this knowledge, we investigated the relationship between SBP conformational dynamics, SBP-ligand interactions and substrate transport.

The general view of SBP conformational changes serving as a binary switch to communicate transport competency may hold for some SBPs, such as OppA (*Figure 2—figure supplement 3B*), while others employ multiple distinct ligand-bound conformations (*Figure 2D–F*; *Figure 4D–G*). To our knowledge, such extreme conformational plasticity of SBPs has not been observed before. MalE shows a remarkable structural flexibility of at least six different ligand-bound conformations (*Figure 2D*; *Figure 4F*). SBD1 (*Figure 2E*; *Figure 4D*) can sample at least four distinct ligand-bound conformations and SBD2 (*Figure 2F*; *Figure 4E*) and OpuAC (*Figure 2B*; *Figure 4G*) at least three. Moreover, MalE, SBD1 and SBD2 have multiple distinct ligand-bound conformations that can all interact with the translocator, as they all facilitate substrate import ('multiple conformations activate

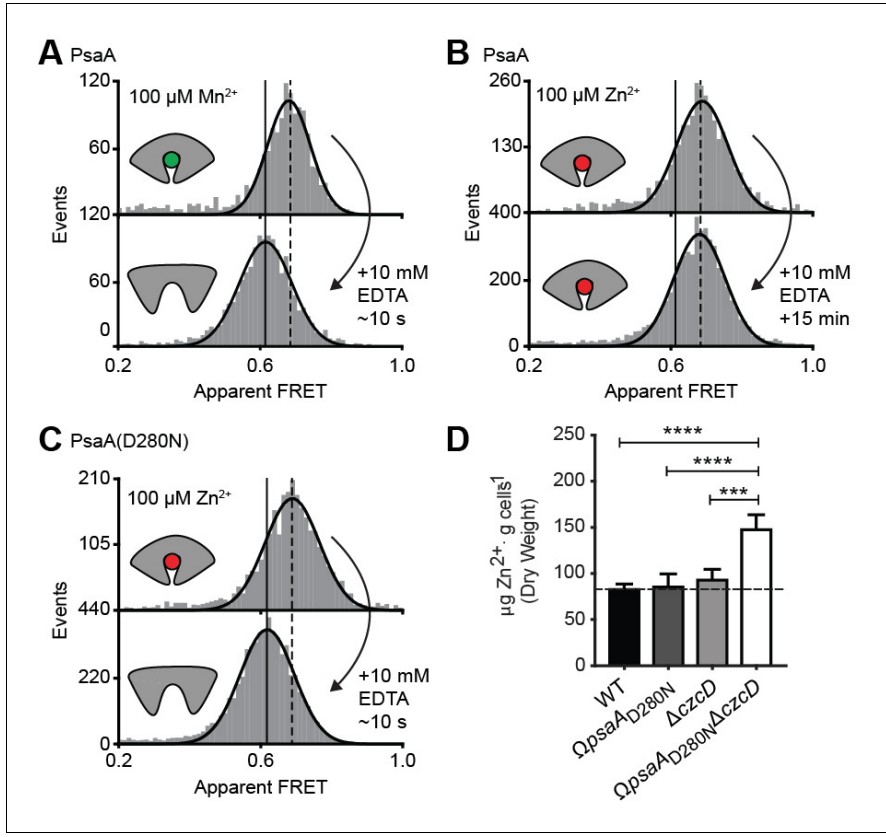

**Figure 5.** Opening transition in PsaA dictates transport specificity. Solution-based apparent FRET efficiency histograms of PsaA(V76C/K237C) in the presence of $Mn^{2+}$ (**A**) or $Zn^{2+}$ (**B**) and PsaA(D280N) in the presence of $Zn^{2+}$ (**C**) upon addition of 10 mM EDTA and incubated for the indicated duration. Bars are the data and the solid line a Gaussian fit. The 95% confidence interval for the mean of the Gaussian distribution can be found in *Supplementary file 3*, and the interval center is indicated by vertical lines (solid, metal-free and dashed, metal-bound). (**D**) Whole cell $Zn^{2+}$ accumulation of *S. pneumoniae* D39 and mutant strains in CDM supplemented with 50 µM $ZnSO_4$ as determined by ICP-MS. Data correspond to mean ± s.d. µg $Zn^{2+}$.$g^{-1}$ dry cell weight from three independent biological experiments. Statistical significance was determined by one-way ANOVA with Tukey post-test (***$p < 0.005$ and ****$p < 0.0001$).
DOI: https://doi.org/10.7554/eLife.44652.021

transport' in *Figure 7*; *Figure 2D–F*). Thus, a productive SBP-translocator interaction in Type I ABC importers can be accomplished without relying on strict structural requirements for docking. This generalization may not apply to all Type I ABC importers since in the Opp importer the translocator might only interact with a unique closed conformation of the SBP OppA (*Figure 2—figure supplement 3B*), and Opp has no measurable affinity for its open ligand-free conformation (*Doeven et al., 2008*).

Exclusion of non-cognate substrates is also a critical biological function for SBPs. Our work has uncovered a hitherto unappreciated complexity in protein-ligand interactions and how this is coupled to regulation of substrate import. Similar to transport, exclusion of non-cognate ligands might be achieved by multiple distinct mechanisms. We have shown that although multiple SBP conformations can activate transport (*Figure 2D–F*), not all SBP conformational states appear to provide the signal to facilitate transport. For example, the binding of certain non-cognate ligands induces a conformational change in SBD1 (*Figure 4D*), MalE (*Figure 4F*) and OpuAC (*Figure 4G*) that are distinct from those that facilitate transport. However, non-cognate substrate binding is not always coupled to an SBP conformational change, as observed for the binding of arginine or lysine to SBD1 and arginine to SBD2 (*Figure 4D–E*). These observations provide a general explanation on how substrate import can fail in Type I ABC importers, which would be due to the SBP-ligand complex assuming a conformation that cannot initiate allosteric interactions with the translocator ('conformational mismatch' in *Figure 7*). A similar hypothesis was put forward based on the observation that binding of β-cyclodextrin fails to fully close MalE (*Hall et al., 1997b*; *Sharff et al., 1993*; *Skrynnikov et al., 2000*). However, the sole observation of partial closing of MalE cannot explain why transport of β-cyclodextrin fails, as we here show that also cognate maltodextrins are able to induce partial closing of MalE (*Figure 2D*).

By contrast, in the $Mn^{2+}$ transporter PsaBCA, a different mechanism is used. In PsaA, the binding site composition of the SBP precludes the ability of the protein to exclude the non-cognate substrate $Zn^{2+}$ from interacting. As a consequence, both metals bind and trigger formation of similar PsaA conformations ('conformational match' in *Figure 7*; *Figure 4H*) (*McDevitt et al., 2011*; *Lawrence et al., 1998*). Despite this, the two ions have starkly different conformational dynamics, with $Zn^{2+}$ forming a highly stable closed conformation, such that it cannot open and release the substrate to its translocator ('SBP cannot open' in *Figure 7*; *Figure 5*). By altering the binding site interactions between PsaA and $Zn^{2+}$, opening is faster and transport of the metal ion can occur (*Figure 5B–D*). Similar observations were made for GlnPQ (*Gouridis et al., 2015*; *Schuurman-Wolters et al., 2018*) and MalE (*Figure 6E*, *Figure 6—figure supplement 3A*), in which a slower/faster opening of the SBP resulted in a decrease/increase in the corresponding transport of the substrate or ATP hydrolysis rate ('faster SBP opening – faster transport' in *Figure 7*). We therefore conclude that for ligands that induce highly stabilized SBP-substrate conformations, which require more energy (thermal or ATP-dependent) to open, transport becomes slower or is abrogated. Based on these findings, we infer that biological selectivity in ABC importers is largely achieved via a combination of ligand release kinetics and its influence on the conformational state of the SBP. This provides a mechanism to facilitate the import of selective substrates, while excluding other compounds. However, our data also implicate a role for the translocator in contributing to the specificity of ABC importers ('rejected by translocator' in *Figure 7*), consistent with previous studies (*Oldham et al., 2013*; *Yu et al., 2015*; *Davidson et al., 1992*; *Speiser and Ames, 1991*).

The presence of a substrate binding site in the translocator of the maltose system is well-established (*Oldham et al., 2013*), although its role, if any, in influencing the rate of transport of maltodextrins is yet unknown. The average time required for the different maltodextrin-MalE complexes to open, correlates positively with the transport and ATP hydrolysis rate (*Figure 6H*) (*Hall et al., 1997a*). This suggests that the substrate, after it has been transferred from MalE to the translocator, acts as a trigger for subsequent steps, for example, the transition from the outward- to the inward-facing transporter conformation or the stimulation of ATP hydrolysis and/or $P_i$ and ADP release ('kinetics of downstream steps are substrate-dependent' in *Figure 7*). Irrespective of the precise molecular mechanism, the positive correlation between lifetime of the SBP closed state and activity of the transporter implies that some maltodextrins trigger certain steps more efficient than other maltodextrins, thereby overcoming the slower opening of MalE, and leading to a preferred uptake of certain maltodextrins over others. When transport is solely altered by changing the SBP conformational dynamics, for example in MalE(I329W/A96W) and PsaA(D280N), the kinetics of these steps

are not affected, as the same ligands are involved, thus explaining the negative correlation between SBP lifetime and transport in these specific cases.

The volume of the binding cavities in the translocator could be a limiting factor for transport via ABC importers. Analysis of the large non-cognate ligands maltooctaose and maltodecaose shows that these are bound reversibly by MalE (*Figure 6A*) and induce conformations similar to that of the cognate ligand maltoheptaose ('conformational match' in *Figure 7*; *Figure 4F*). Therefore, failure of the maltose system to transport maltooctaose and maltodecaose most likely arises due to size

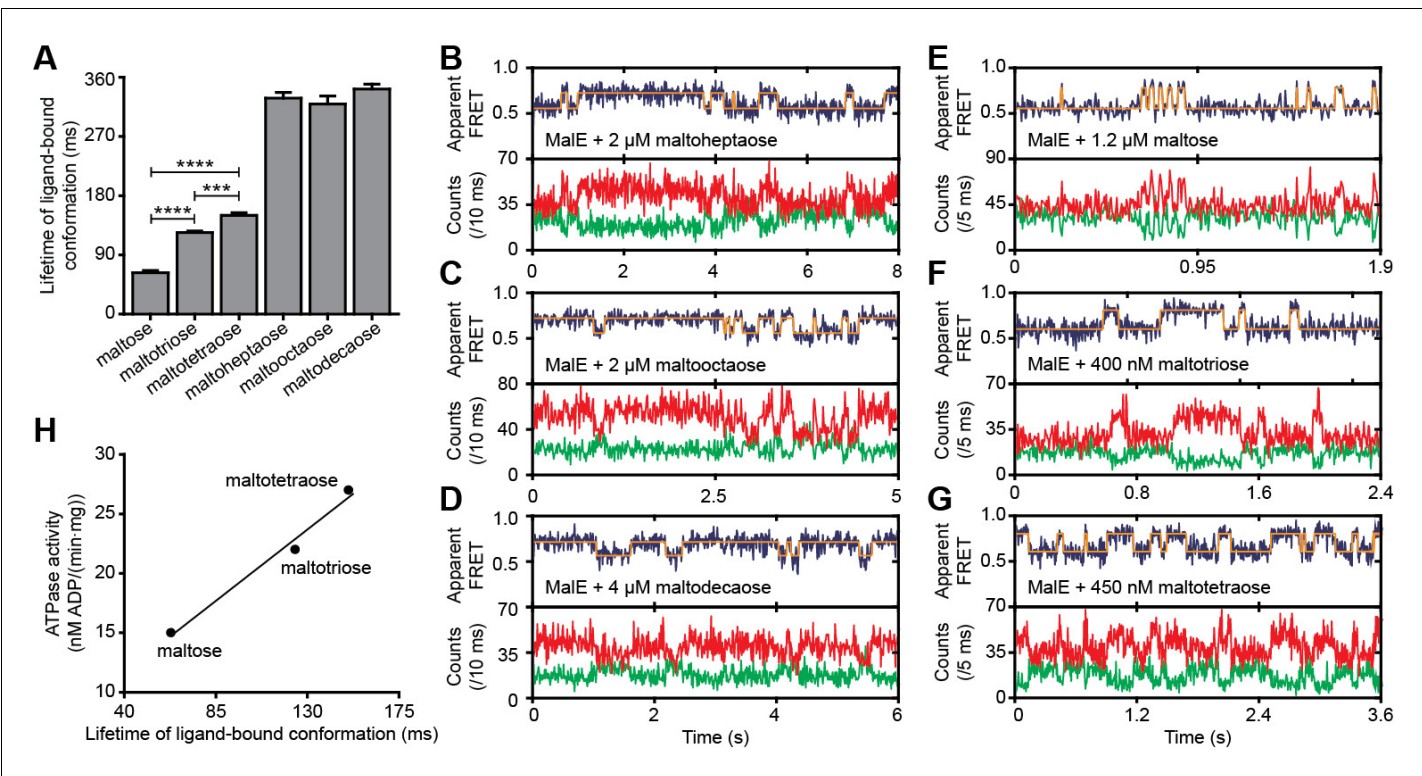

**Figure 6.** Lifetime of MalE ligand-bound conformations and relation to activity. (**A**) Mean lifetime of the ligand-bound conformations of MalE, obtained from all single-molecule fluorescence trajectories in the presence of different maltodextrins as indicated. Data corresponds to mean ± s.e.m. Data in *Figure 6—figure supplement 2*. Statistical significance was determined by two-tailed unpaired *t*-tests (***p < 0.005 and ****p < 0.0001). (**B, C, D, E, F and G**) Representative fluorescence trajectories of MalE(T36C/S352C) in the presence of different substrates as indicated. In all fluorescence trajectories presented: top panel shows calculated apparent FRET efficiency (blue) from the donor (green) and acceptor (red) photon counts as shown in the bottom panels. Most probable state-trajectory of the Hidden Markov Model (HMM) is shown (orange). (**H**) Published ATPase activity (*Hall et al., 1997a*) linked to the lifetime of the closed MalE conformation induced by transport of different cognate substrates as indicated. Points are the data and the solid line a simple linear regression fit.

DOI: https://doi.org/10.7554/eLife.44652.022

The following source data and figure supplements are available for figure 6:

**Source data 1.** Lifetimes of the high FRET state of the data shown in *Figure 6A* and *Figure 6—figure supplement 2*.
DOI: https://doi.org/10.7554/eLife.44652.026

**Source data 2.** Donor and acceptor photon counts, apparent FRET efficiency and most probable state-trajectory of the Hidden Markov Model of the traces in *Figure 6B–G*.
DOI: https://doi.org/10.7554/eLife.44652.027

**Source data 3.** Lifetimes of the high FRET state of the data shown in *Figure 6—figure supplement 3B*.
DOI: https://doi.org/10.7554/eLife.44652.028

**Figure supplement 1.** Surface-based smFRET histogram of MalE.
DOI: https://doi.org/10.7554/eLife.44652.023

**Figure supplement 2.** Lifetime distribution of the ligand-bound conformations of MalE.
DOI: https://doi.org/10.7554/eLife.44652.024

**Figure supplement 3.** Conformational changes and dynamics of MalE(A96W/I329W).
DOI: https://doi.org/10.7554/eLife.44652.025

limitations of the translocator rather than failure of MalE to close and release the bound ligand. This supposition is supported by an analysis of the binding cavities in the crystal structure of MalFGK$_2$-MalE (*Oldham et al., 2013*). These data suggest that the transporter could only accommodate maltodextrins as large as maltoheptoase. In contrast, MalE could accommodate larger maltodextrins, including β-cyclodextrin (*Figure 4F*), probably due to its greater structural flexibility (*Figure 2D*; *Figure 4F*), thereby allowing the binding pocket to adapt and ligands to extend into the solvent phase.

The presence of two consecutive binding pockets, one in the SBP and one in the translocator, in at least some ABC importers could indicate that specificity of transport occurs through a proofreading mechanism in a manner analogous to aminoacyl-tRNA synthetases and DNA polymerase (*Shevelev and Hübscher, 2002*; *Kotik-Kogan et al., 2005*). In such a mechanism, a substrate can be rejected even if it has been bound by the SBP. Although we show that intrinsic closing is a rare event ('little intrinsic closing' in *Figure 7*; data in *Figure 3*), it might influence transport in a cellular context where the ratio between SBP and translocator can be high (*Schmidt et al., 2016*). Moreover, other fast (μs-ms) and short-range conformational changes might be present as shown by NMR analysis on MalE (*Tang et al., 2007*). We speculate that in Type I ABC importers the wasteful conversion of chemical energy is prevented by a proofreading mechanism, as any thermally driven closing event would not be able to initiate the translocation cycle, as the substrate is absent. In accordance, ATP hydrolysis and transport are tightly coupled in the Type I importer GlnPQ (*Lycklama A Nijeholt et al., 2018*) that, based on the crystal structure of the homologous Art(QM)$_2$ (*Yu et al., 2015*), contains an internal binding pocket located within the TMDs. By contrast, futile hydrolysis of ATP in the Type II importer BtuCDF (*Borths et al., 2005*) appears to correlate with the lack of a defined binding pocket inside the TMDs.

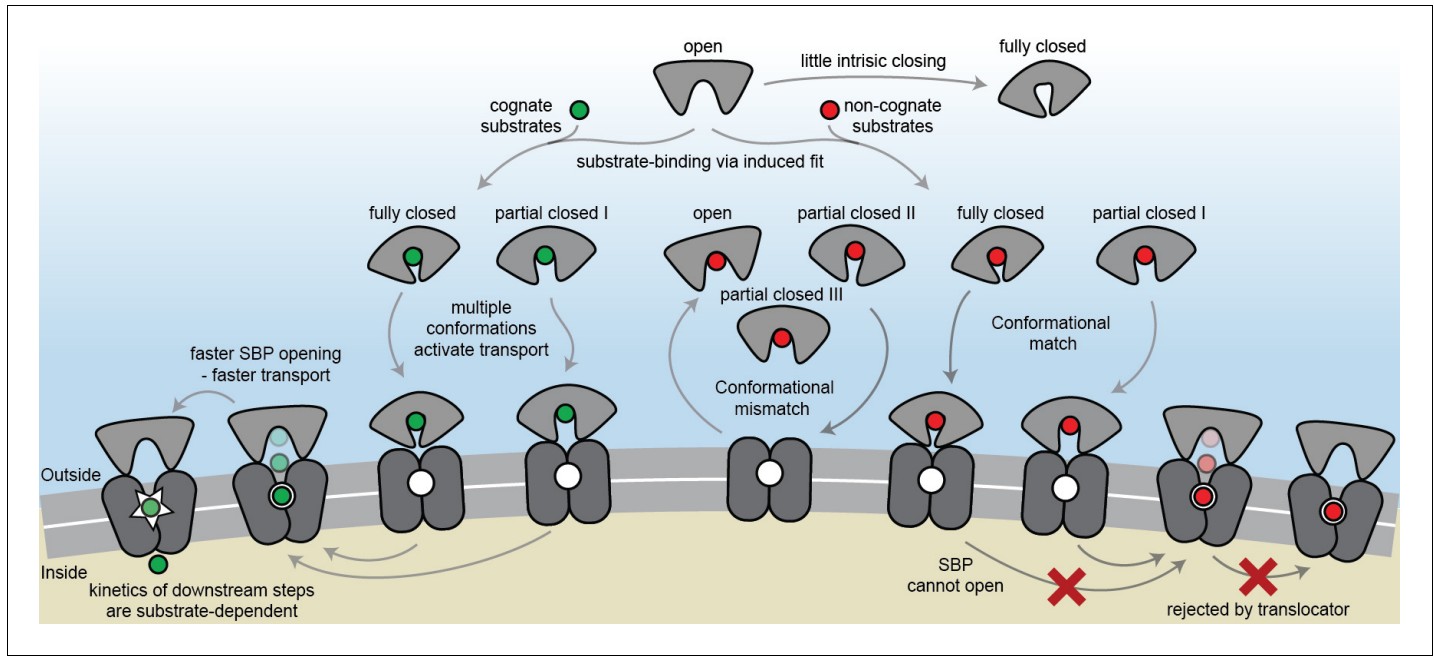

**Figure 7.** The conformational changes and dynamics of SBPs and the regulation of transport. Schematic summarizing the plasticity of ligand binding and solute import via ABC importers. Intrinsic closing of an SBP is a rare event or absent in some SBPs ('little intrinsic closing'). Ligands are bound via induced fit ('ligand-binding via induced fit'). SBPs can acquire one or more conformations that can activate transport ('multiple conformations activate transport'). Variations in cognate substrate transport are caused by: (i) openings rate of the SBP and substrate transfer to the translocator ('faster SBP opening – faster transport') and (ii) substrate-dependent downstream steps ('kinetics of downstream steps are substrate-dependent'). Although SBPs can acquire a conformation that activates transport ('conformational match'), transport still fails when: (i) the SBP has no affinity for the translocator and/or cannot make the allosteric interaction with the translocator ('conformational mismatch'); (ii) the SBP cannot open and release the substrate to the translocator ('SBP cannot open'); or (iii) due to the specificity and size limitations of the translocator ('rejected by translocator').
DOI: https://doi.org/10.7554/eLife.44652.029

# Materials and methods

## Key resources table

| Reagent type (species) or resource | Designation | Source or reference | Identifiers | Additional information |
|---|---|---|---|---|
| Gene (*Escherichia coli*) | *MalE* | NA | UniProt: P0AEX9 | |
| Antibody | Mouse anti-his | Qiagen | RRID:AB_2714179 | (1:200) |
| Strain, strain background (*Streptococcus pneumoniae*) | D39 | National Collection of Type Cultures | NCTC:7466 | Capsular serotype 2 |
| Strain, strain background (*Streptococcus pneumoniae*) | D39 Δ*psaA* | This paper | | Replacement of *psaA* with the Janus cassette (Δ*psaA*::*Janus*) |
| Strain, strain background (*Streptococcus pneumoniae*) | D39 Δ*czcD* | This paper | | Replacement of *czcD* with the Janus cassette (Δ*czcD*::*Janus*) |
| Strain, strain background (*Streptococcus pneumoniae*) | D39 Ω*psaA*$_{D280N}$ | This paper | | Replacement of Δ*psaA*::*Janus* with *psaA* D280N (Δ*psaA*::*psaA*$_{D280N}$) |
| Strain, strain background (*Streptococcus pneumoniae*) | D39 Ω*psaA*$_{D280N}$Δ*czcD* | This paper | | Replacement of Δ*psaA*::*Janus* with *psaA* D280N; replacement of *czcD* with the Janus cassette (Δ*psaA*::*psaA*$_{D280N}$Δ*czcD*::*Janus*) |
| Strain, strain background (*Lactococcus lactis*) | NZ9000 | NIZO food research | | |
| Strain, strain background (*Lactococcus lactis*) | GKW9000 | DOI: 10.1038/nsmb2929 | | Lactococcus lactis NZ9000 with *glnPQ* gene deleted |
| Strain, strain background (*Escherichia coli*) | K12 | Other | | Provided by Tassos Economou, KU Leuven |
| Strain, strain background (*Escherichia coli*) | BL 21 DE3 | Other | | Provided by Tassos Economou, KU Leuven |
| Recombinant DNA reagent | pET20b | Merck | Cat#:69739–3 | |
| Recombinant DNA reagent | pNZglnPQhis | DOI: 10.1047/jbc.M500522200 | | Expression plasmid for GlnPQ |
| Recombinant DNA reagent | SBD1-T159C/G87C | DOI: 10.1038/nsmb2929 | | Expression plasmid for SBD1(T159C/G87C) |
| Recombinant DNA reagent | SBD2-T369C/S451C | DOI: 10.1038/nsmb2929 | | Expression plasmid for SBD2(T369C/S451C) |
| Recombinant DNA reagent | pCAMcLIC01-PsaA | DOI: 10.1038/nchembio.1382 | | Expression plasmid for PsaA |
| Recombinant DNA reagent | pCAMcLIC01-PsaAD280N | DOI: 10.1038/nchembio.1382 | | Expression plasmid for PsaA(D280N) |
| Recombinant DNA reagent | pNZOpuCHis | DOI: 10.1093/emboj/cdg581 | | Expression plasmid for OpuAC |

*Continued on next page*

*Continued*

| Reagent type (species) or resource | Designation | Source or reference | Identifiers | Additional information |
|---|---|---|---|---|
| Recombinant DNA reagent | pNZcLIC-OppA | DOI: 10.1002/pro.97 | | Expression plasmid for OppA |
| Recombinant DNA reagent | PsaA-V76C/K237C | This paper | | Expression plasmid for PsaA(V76C/K237C) from the pCAMcLIC01-PsaA construct |
| Recombinant DNA reagent | PsaA-E74C/K237C | This paper | | Expression plasmid for PsaA(E74C/K237C) from the pCAMcLIC01-PsaA construct |
| Recombinant DNA reagent | PsaA-D280N/V76C/K237C | This paper | | Expression plasmid for PsaA(D280N/V76C/K237C) from the pCAMcLIC01-PsaAD280N construct |
| Recombinant DNA reagent | MalE-T36C/S352C | This paper | | Progenitors: PCR, *E. coli* gDNA; pET20b vector |
| Recombinant DNA reagent | MalE-T36C/N205C | This paper | | Progenitors: PCR, *E. coli* gDNA; pET20b vector |
| Recombinant DNA reagent | MalE-K34C/R354C | This paper | | Progenitors: PCR, *E. coli* gDNA; pET20b vector |
| Recombinant DNA reagent | MalE-T36C/S352C/A96W/I329W | This paper | | Progenitors: PCR, *E. coli* gDNA; pET20b vector |
| Recombinant DNA reagent | OpuAC-V360C/N423C | This paper | | Expression plasmid for OpuAC(V360C/N423C) from the pNZOpuCHis construct |
| Recombinant DNA reagent | OppA-A209C/S441C | This paper | | Expression plasmid for OppA(A209C/S441C) from the pNZcLIC-OppA construct |
| Sequence-based reagent | Primers | Merck | | see *Supplementary File 2* |
| Peptide, recombinant protein | RPPGFSPFR | Merck | Cat#:B3259 | peptide sequence: RPPGFSPFR |
| Peptide, recombinant protein | RDMPIQAF | CASLO ApS | | peptide sequence: RDMPIQAF |
| Peptide, recombinant protein | SLSQSKVLPVPQ | CASLO ApS | | peptide sequence: SLSQSKVLPVPQ |
| Peptide, recombinant protein | SLSQSKVLP | CASLO ApS | | peptide sequence: SLSQSKVLP |
| Chemical compound, drug | Glycine Betaine | Merck | Cat#:B3501 | |
| Chemical compound, drug | Carnitine | Merck | Cat#:94954 | |

*Continued on next page*

*Continued*

| Reagent type (species) or resource | Designation | Source or reference | Identifiers | Additional information |
|---|---|---|---|---|
| Chemical compound, drug | Maltose | Merck | Cat#:63418 | |
| Chemical compound, drug | Maltotriose | Merck | Cat#:851493 | |
| Chemical compound, drug | Maltotetraose | Carbosynth Limited | Cat#:OM06979 | |
| Chemical compound, drug | Maltopentaose | Merck | Cat#:M8128 | |
| Chemical compound, drug | Maltohexaose | Santa Cruz Biotechnology | Cat#:sc-218665 | |
| Chemical compound, drug | Maltoheptaose | Carbosynth Limited | Cat#:OM06868 | |
| Chemical compound, drug | Maltodecaose | Carbosynth Limited | Cat#:OM146832 | |
| Chemical compound, drug | Maltooctaose | Carbosynth Limited | Cat#:OM06941 | |
| Chemical compound, drug | Beta Cyclodextrin | Merck | Cat#:C4767 | |
| Chemical compound, drug | Maltotetroitol | Carbosynth Limited | Cat#:OM02796 | |
| Chemical compound, drug | Maltotriitol | Merck | Cat#:M4295 | |
| Chemical compound, drug | $^3$H-Asparagine | American Radiolabeled Chemicals | Cat#:ART 0500–250 µCi | |
| Chemical compound, drug | $^{14}$C-Glutamine | PerkinEllmer | Cat#: NEC451050UC | |
| Chemical compound, drug | $^{14}$C-Histidine | PerkinEllmer | Cat#: NEC277E050UC | |
| Chemical compound, drug | $^{14}$C-Arginine | Moravek | Cat#:MC 137 | |
| Chemical compound, drug | $^3$H-Lysine | PerkinEllmer | Cat#: NET376250UC | |
| Chemical compound, drug | Alexa555 | Thermo Fisher Scientific | Cat#:A20346 | |
| Chemical compound, drug | Alexa647 | Thermo Fisher Scientific | Cat#:A20347 | |

*Continued on next page*

*Continued*

| Reagent type (species) or resource | Designation | Source or reference | Identifiers | Additional information |
|---|---|---|---|---|
| Chemical compound, drug | Cy3B | GE Healthcare | Cat#:PA63131 | |
| Chemical compound, drug | ATTO647N | ATTO-TECH | Cat#: AD 647 N-45 | |
| Software, algorithm | Dual-Channel-Burst-Search | DOI: 10.1021/jp063483n | | |
| Software, algorithm | LabView data acquisition | DOI: 10.1371/journal.pone.0175766 | | Provided by Shimon Weiss, UCLA |
| Software, algorithm | Hidden Markov Model | DOI: 10.1109/5.18626 | | |
| Software, algorithm | Origin | OriginLab | RRID: SCR_002815 | |
| Software, algorithm | MATLAB | MathWorks | RRID: SCR_001622 | |

## Gene expression and SBP purification

N-terminal extension of the soluble SBPs with a His$_x$ tag (His$_{10}$PsaA, His$_{10}$SBD1, His$_{10}$SBD2, His$_{10}$OppA and His$_6$OpuAC) were expressed and purified as previously described (*Gouridis et al., 2015*; *Wolters et al., 2010*; *Doeven et al., 2004*; *Couñago et al., 2014*). Protein derivatives having the cysteine point mutations were constructed using QuickChange mutagenesis (*Bok and Keller, 2012*) or Megaprimer PCR mutagenesis (*Vander Kooi, 2013*) protocols. Primers are indicated in *Supplementary file 2* and all sequences were by sequencing. OppA, OpuAC, PsaA and PsaA (D280N) derivatives were constructed using as templates vectors pNZcLIC-OppA (*Berntsson et al., 2009*), pNZOpuCHis (*Biemans-Oldehinkel and Poolman, 2003*), pCAMcLIC01-PsaA (*Couñago et al., 2014*) and pCAMcLIC01-PsaAD280N (*Couñago et al., 2014*), respectively. Construction of SBD1 and SBD2 cysteine derivatives was accomplished as described previously (*Gouridis et al., 2015*).

The *mal*E gene (UniProt: P0AEX9) was isolated from the genome of *Escherichia coli* K12. The primers were designed to exclude the signal peptide (amino acids 1–26). Primers introduced *Nde*I and *Hind*III restriction sites, and the gene product was sub-cloned in the pET20b vector (Merck). MalE derivatives having the cysteine or other point mutations were constructed using QuickChange mutagenesis (*Bok and Keller, 2012*) and Megaprimer PCR mutagenesis (*Vander Kooi, 2013*) protocols. Primers are indicated in *Supplementary file 2* and all sequences were verified by sequencing. His$_6$-MalE was over-expressed in *E. coli* BL21 DE3 cells (*F–ompT gal dcm lon hsdSB($r_B$–$m_B$) λ(DE3 [lacI lacUV5-T7p07 ind1 sam7 nin5]) [malB+]K-12(λS)*). Cells harbouring plasmids expressing the MalE wild-type and derivatives were grown at 30°C until an optical density (OD$_{600}$) of 0.5 was reached. Protein expression was then induced by addition of 0.25 mM isopropyl β-D-1-thiogalactopyranoside (IPTG). After 2 hr induction cells were harvested. DNase 500 ug/ml (Merck) was added and passed twice through a French pressure cell at 1,500 psi and 2 mM phenylmethylsulfonyl fluoride (PMSF) was added to inhibit proteases. The soluble supernatant was isolated by centrifugation at 50,000 $\times$ g for 30 min at 4°C. The soluble material was then purified and loaded on Ni$^{2+}$-sepharose resin (GE Healthcare) in 50 mM Tris-HCl, pH 8.0, 1 M KCl, 10% glycerol, 10 mM imidazole and 1 mM dithiothreitol (DTT; Sigma-Aldrich). The immobilized proteins were washed (50 mM Tris-HCl, pH 8.0, 50 mM KCl, 10% glycerol, 10 mM imidazole and 1 mM DTT plus 50 mM Tris-HCl, pH 8.0, 1 M KCl, 10% glycerol, 30 mM imidazole and 1 mM DTT sequentially) and then eluted (50 mM Tris-HCl, pH 8.0, 50 mM KCl, 10% glycerol, 300 mM imidazole and 1 mM DTT). Protein fractions were pooled (supplemented with 5 mM EDTA and 10 mM DTT), concentrated (10.000 MWCO Amicon; Merck-Millipore), dialyzed against 100–1000 volumes of buffer (50 mM Tris-HCl, pH 8.0, 50 mM KCl, 50% glycerol and 10 mM DTT), aliquoted and stored at −20°C until required.

## Uptake experiments in whole cells

*Lactococcus lactis* GKW9000 carrying pNZglnPQhis (*Schuurman-Wolters and Poolman, 2005*) was cultivated semi-anaerobically at 30°C in M17 (Oxoid) medium supplemented with 1% (w/v) glucose and 5 µg/ml chloramphenicol. For uptake experiments cells were grown in GM17 to an $OD_{600}$ of 0.4, induced for 1 hr with 0.01% of culture supernatant of the nisin A-producing strain NZ9700 and harvested by centrifugation for 10 min at 4000 x *g*; the final nisin A concentration is ~1 ng/ml. After washing twice with 10 mM PIPES-KOH, 80 mM KCl, pH 6.0, the cells were resuspended to $OD_{600} = 50$ in the same buffer. Uptake experiments were performed at 0.1–0.5 mg/ml total protein in 30 mM PIPES-KOH, 30 mM MES-KOH, 30 mM HEPES-KOH (pH 6.0). Before starting the transport assays, the cells were equilibrated and energized at 30°C for 3 min in the presence of 10 mM glucose plus 5 mM $MgCl_2$. After 3 min, the uptake reaction was started by addition of either $[^{14}C]$-glutamine, $[^{14}C]$-histidine, $[^{14}C]$-lysine (all from PerkinElmer), $[^{14}C]$-arginine (Moravek) or $[^3H]$-asparagine (ARC); the specific radioactivity was adjusted for each experiment (amino-acid concentration) to obtain sufficient signal above background; the final amino acid concentrations are indicated in the figure legends. At given time intervals, samples were taken and diluted into 2 ml ice-cold 100 mM LiCl. The samples were rapidly filtered through 0.45 µm pore-size cellulose nitrate filters (Amersham) and the filter was washed once with ice-cold 100 mM LiCl. The radioactivity on the filters was determined by liquid scintillation counting.

## Purification and membrane reconstitution of GlnPQ for in vitro transport assays

Membrane vesicles of *Lactococcus lactis* GKW9000 carrying pNZglnPQhis (*Schuurman-Wolters and Poolman, 2005*) were prepared as described before (*Lycklama A Nijeholt et al., 2018*). For reconstitution into proteoliposomes, 150 mg of total protein in membrane vesicles was solubilized in 50 mM potassium phosphate pH 8.0, 200 mM NaCl, 20% glycerol and 0.5% (w/v) DDM for 30 min at 4°C. The sample was centrifuged (12 min, 300,000x*g*) and the supernatant was collected. Subsequently, GlnPQ was allowed to bind to Ni-Sepharose (1.5 ml bed volume) for 1 hr at 4°C after addition of 10 mM imidazole. The resin was rinsed with 20 column volumes of wash buffer (50 mM potassium phosphate, pH 8.0, 200 mM NaCl, 20% (v/v) glycerol, 50 mM imidazole and 0.02% (w/v) DDM). The protein was eluted with five column volumes of elution buffer (50 mM potassium phosphate, pH 8.0, 200 mM NaCl, 10% (w/v) glycerol, 500 mM imidazole plus 0.02% (w/v) DDM). The purified GlnPQ was used for reconstitution into liposomes composed of egg yolk L-α-phosphatidylcholine and purified *E. coli* lipids (Avanti polar lipids) in a 1:3 ratio (w/w) as described before (*Geertsma et al., 2008*) with a final protein/lipid ratio of 1:100 (w/w). An ATP regenerating system, consisting of 50 mM potassium phosphate, pH 7.0, creatine kinase (2.4 mg/ml), $Na_2$-ATP (10 mM), $MgSO_4$ (10 mM), and $Na_2$-creatine-phosphate (24 mM) was enclosed in the proteoliposomes by two freeze/thaw cycles, after which the vesicles were stored at −80°C. On the day of the uptake experiment, the proteoliposomes were extruded 13 times through a polycarbonate filter (200 nm pore size), diluted to 3 ml with 100 mM potassium phosphate, pH 7.0, centrifuged (265,000 g for 20 min), and then washed and resuspended in 100 mM potassium phosphate, pH 7.0, to a concentration of 50 mg of lipid/ml.

Uptake in proteoliposomes was measured in 100 mM potassium phosphate, pH 7.0, supplemented with 5 µM of $[^{14}C]$-glutamine or $[^3H]$-asparagine. This medium, supplemented with or without unlabeled amino acids (asparagine, arginine, glutamine, histidine or lysine), was incubated at 30°C for 2 min prior to adding proteoliposomes (kept on ice) to a final concentration of 1–5 mg of lipid/ml. At given time intervals, 40 µl samples were taken and diluted with 2 ml of ice-cold isotonic buffer (100 mM potassium phosphate, pH 7.0). The samples were collected on 0.45 m pore size cellulose nitrate filters and washed twice as described above. After addition of 2 ml Ultima Gold scintillation liquid (PerkinElmer), radioactivity was measured on a Tri-Carb 2800TR (PerkinElmer). A single time-dependent uptake experiment is shown in *Figure 4A–C* and consistent results were obtained upon repetition with an independent sample preparation.

## Zinc accumulation in whole cells

The *S. pneumoniae* D39 mutant strains $\Omega psaA_{D280N}$ and $\Delta czcD$ were constructed using the Janus cassette system (*Sung et al., 2001*). Briefly, the upstream and downstream flanking regions of *psaA*

and *czcD* were amplified using primers (**Supplementary file 2**) with complementarity to either *psaA*$_{D280N}$ ($\Omega psaA_{D280N}$), generated via site-directed mutagenesis of *psaA* following manufacturer instructions (Agilent), or the Janus cassette ($\Delta czcD$) and were joined by overlap extension PCR. These linear fragments were used to replace by homologous recombination *psaA* and *czcD*, respectively, in the chromosome of wild-type and $\Delta czcD$ strains. For metal accumulation analyses, *S. pneumoniae* strains were grown in a cation-defined semi-synthetic medium (CDM) with casein hydrolysate and 0.5% yeast extract, as described previously (**Plumptre et al., 2014**). Whole cell metal ion accumulation was determined by inductively coupled plasma-mass spectrometry (ICP-MS) essentially as previously described (**Begg et al., 2015**). Briefly, *S. pneumoniae* strains were inoculated into CDM supplemented with 50 μM ZnSO$_4$ at a starting OD$_{600}$ of 0.05 and grown to mid-log phase (OD$_{600}$ = 0.3–0.4) at 37°C in the presence of 5% CO$_2$. Cells were washed by centrifugation six times in PBS with 5 mM EDTA, harvested, and desiccated at 95°C for 18 hr. Metal ion content was released by treatment with 500 μL of 35% HNO$_3$ at 95°C for 60 min. Metal content was analysed on an Agilent 8900 QQQ ICP-MS (**Couñago et al., 2014**).

## Isothermal titration calorimetry (ITC)

Purified OppA was dialyzed overnight against 50 mM Tris-HCl, pH 7.4, 50 mM KCl. ITC experiments were carried by microcalorimetry on a ITC200 calorimeter (MicroCal). The peptide (RPPGFSFR) stock solution (200 μM) was prepared in the dialysis buffer and was stepwise injected (2 μl) into the reaction cell containing 20 μM OppA. All experiments were carried out at 25°C with a mixing rate of 400 rpm. Data were analyzed with a one site-binding model using, provided by the Origin software (OriginLab).

## Protein labeling for FRET measurements

Surface-exposed and non-conserved positions were chosen for Cys engineering and subsequent labeling, based on X-ray crystal structures of OpuAC (3L6G, 3L6H), SBD1 (4AL9), SBD2 (4KR5, 4KQP), PsaA (3ZK7, 1PSZ), OppA (3FTO, 3RYA) and MalE (1OMP, 1ANF). Unlabeled protein derivatives (20–40 mg/ml) were stored at −20°C in the appropriate buffer (50 mM Tris-HCl, pH 7.4, 50 mM KCl, 50% glycerol for MalE and OppA. 25 mM Tris-HCl, pH 8.0, 150 mM NaCl, 1 μM EDTA, 50% glycerol for PsaA. 50 mM KPi, pH 7.4, 50 mM KCl, 50% glycerol for OpuAC, SBD1 and SBD2) supplemented with 1 mM DTT.

Stochastic labeling was performed with the maleimide derivative of dyes Cy3B (GE Healthcare) and ATTO647N (ATTO-TEC) for OpuAC. MalE, SBD1, SBD2, OppA and PsaA were labeled with Alexa555 and Alexa647 maleimide (ThermoFisher). The purified proteins were first treated with 10 mM DTT for 30 min to reduce oxidized cysteines. After dilution of the protein sample to a DTT concentration of 1 mM the reduced protein were immobilized on a Ni$^{2+}$-Sepharose resin (GE Healthcare) and washed with 10 column volumes of buffer A (50 mM Tris-HCl, pH 7.4, 50 mM KCl for MalE and OppA. 25 mM Tris-HCl, pH 8.0, 150 mM NaCl, 1 μM EDTA for PsaA. 50 mM KPi, pH 7.4, 50 mM KCl for OpuAC, SBD1 and SBD2) to remove the DTT. To make sure that no endogenous ligand was left, for some experiments, and prior to removing the DTT, we unfolded the immobilized-SBPs by treatment with 6 M of urea supplemented with 1 mM DTT and refolded them again by washing with buffer A. The resin was incubated 1–8 hr at 4°C with the dyes dissolved in buffer A. To ensure a high labeling efficiency, the dye concentration was ~20 times higher than the protein concentration. Subsequently, unbound dyes were removed by washing the column with at least 20 column volumes of buffer A. Elution of the proteins was done by supplementing buffer A with 400 mM imidazole. The labeled proteins were further purified by size-exclusion chromatography (Superdex 200, GE Healthcare) using buffer A. Sample composition was assessed by recording the absorbance at 280 nm (protein), 559 nm (donor), and 645 nm (acceptor) to estimate labeling efficiency. For all proteins, the labeling efficiency was >90%.

## Fluorescence anisotropy

To verify that the measurements of apparent FRET efficiency report on inter-probe distances between the donor and acceptor fluorophores, at least one of the fluorophores must be able to rotate freely. To investigate this, we determined the anisotropy values of labeled proteins. The fluorescence intensity was measured on a scanning spectrofluorometer (Jasco FP-8300; 10 nm excitation

and emission bandwidth; 8 s integration time) around the emission maxima of the fluorophores (for donor, $\lambda_{ex}$ = 535 nm and $\lambda_{em}$ = 580 nm; for acceptor, $\lambda_{ex}$ = 635 nm and $\lambda_{em}$ = 660 nm). Anisotropy values $r$ were obtained from on $r = (I_{VV} - GI_{VH})/(I_{VV} + 2GI_{VH})$, where $I_{VV}$ and $I_{VH}$ are the fluorescence emission intensities in the vertical and horizontal orientation, respectively, upon excitation along the vertical orientation. The sensitivity of the spectrometer to different polarizations was corrected via the factor $G = I_{HV}/I_{HH}$, where $I_{HV}$ and $I_{HH}$ are the fluorescence emission intensities in the vertical and horizontal orientation, respectively, upon excitation along the horizontal orientation. $G$-values were determined to be 1.8-1.9. The anisotropy was measured in buffer A and the labeled proteins and free-fluorophores in a concentration range of $50-500$ nM at room temperature.

## Solution-based smFRET and ALEX

Solution-based smFRET and alternating laser excitation (ALEX) (*Kapanidis et al., 2004*) experiments were carried out at 25–100 pM of labeled protein at room temperature in buffer A supplemented with additional reagents as stated in the text. Microscope cover slides (no. 1.5H precision cover slides, VWR Marienfeld) were coated with 1 mg/mL BSA for 30–60 s to prevent fluorophore and/or protein interactions with the glass material. Excess BSA was subsequently removed by washing and exchange with buffer A.

All smFRET experiments were performed using a home-built confocal microscope. In brief, two laser-diodes (Coherent Obis) with emission wavelength of 532 and 637 nm were directly modulated for alternating periods of 50 µs and used for confocal excitation. The laser beams where coupled into a single-mode fiber (PM-S405-XP, Thorlabs) and collimated (MB06, Q-Optics/Linos) before entering a water immersion objective (60X, NA 1.2, UPlanSAPO 60XO, Olympus). The fluorescence was collected by excitation at a depth of 20 µm. Average laser powers were 30 µW at 532 nm (~30 kW/cm$^2$) and 15 µW at 637 nm (~15 kW/cm$^2$). Excitation and emission light was separated by a dichroic beam splitter (zt532/642rpc, AHF Analysentechnik), which is mounted in an inverse microscope body (IX71, Olympus). Emitted light was focused onto a 50 µm pinhole and spectrally separated (640DCXR, AHF Analysentechnik) onto two single-photon avalanche diodes (TAU-SPADs-100, Picoquant) with appropriate spectral filtering (donor channel: HC582/75; acceptor channel: Edge Basic 647LP; AHF Analysentechnik). Registration of photon arrival times and alternation of the lasers was controlled by an NI-Card (PXI-6602, National Instruments) using LabView data acquisition software of the Weiss laboratory (*Ingargiola et al., 2017*).

An individual labeled protein diffusing through the confocal volume generates a burst of photons. To identify fluorescence bursts a dual-channel burst search (*Nir et al., 2006*) was used with parameters M = 15, T = 500 µs and L = 25. In brief, a fluorescent signal is considered a burst, when a total of L photons having M neighboring photons within a time window of length T centred on their own arrival time. A first burst search was done that includes the donor and acceptor photons detected during the donor excitation, and a second burst search was done including only the acceptor photons detected during the acceptor excitation. The two separate burst searches were combined to define intervals when both donor and acceptor fluorophores are active. These intervals define the bursts. Only bursts having >150 photons were further analysed

The three relevant photon streams were analysed (DA, donor-based acceptor emission; DD, donor-based donor emission; AA, acceptor-based acceptor emission) and assignment is based on the excitation period and detection channel (*Kapanidis et al., 2004*). The apparent FRET efficiency is calculated via F(DA)/[F(DA)+F(DD)] and the Stoichiometry S by [F(DD)+F(DA)]/[(F(DD)+F(DA)+F(AA)], where F(·) denotes the summing over all photons within the burst (*Kapanidis et al., 2004*). The accurate FRET efficiency E was calculated by correcting the apparent FRET efficiency for background, direct excitation of the acceptor by donor excitation, leakage of donor fluorescence in the acceptor detection channel and relative differences in the efficiencies of the detectors and the quantum yield of the dyes (*Nir et al., 2006*). Corrections are made using established protocols as described in Lee et al (*Nir et al., 2006*). From the average E (see below), the mean inter-dye distance R was calculated via E = $1/(1+(R/R_0)^6)$, using $R_0$ of 5.1 nm for Alexa555/Alexa647 and 6.2 nm for Cy3B/Atto647N.

Binning the detected bursts into a 2D (apparent) FRET/S histogram allowed the selection of the donor and acceptor labeled molecules and reduce artefacts arising from fluorophore bleaching (*Kapanidis et al., 2004*). The selected (apparent) FRET histogram were fitted with a Gaussian distribution using nonlinear least square, to obtain a 95% Wald confidence interval for the distribution

mean. Statements about the significance of the mean of the FRET distributions are based on a comparison of the appropriate confidence intervals. In addition, a two-way Kolmogorov-Smirnov test was performed, as implemented in Matlab (MathWorks), on the selected burst corresponding to donor and acceptor-labeled proteins.

## Scanning confocal microscopy

Confocal scanning experiments were performed at room temperature and using a home-built confocal scanning microscope as described previously (*Husada et al., 2018*). In brief, surface scanning was performed using a XYZ-piezo stage with 100 × 100 × 20 µm range (P-517–3 CD with E-725.3CDA, Physik Instrumente). The detector signal was registered using a HydraHarp 400 picosecond event timer and a module for time-correlated single photon counting (both Picoquant). Data were recorded with constant 532 nm excitation at an intensity of 0.5 µW (~125 W/cm$^2$) for SBD1, SBD2, PsaA, OppA and MalE, but 1.5 µW (~400 W/cm$^2$) for OpuAC. Scanning images of 10 × 10 µm were recorded with 50 nm step size and 2 ms integration time at each pixel. After each surface scan, the positions of labeled proteins were identified manually; the position information was used to subsequently generate time traces. Surface immobilization was conducted using an anti-HIS antibody and established surface-chemistry protocols as described (*Gouridis et al., 2015*). A flow-cell arrangement was used as described before (*Gouridis et al., 2015*; *Roy et al., 2008*) for studies of surface-tethered proteins, except for MalE. MalE was studied on standard functionalized coverslides since MalE was extremely sensitive to contaminations of maltodextrins in double-sided tape or other flow-cell parts. All experiments of OpuAC and PsaA were carried out in degassed buffer A under oxygen-free conditions obtained utilizing an oxygen-scavenging system supplemented with 10 mM of (±)−6-Hydroxy-2,5,7,8-tetramethylchromane-2-carboxylic acid (Trolox; Merck) (*van der Velde et al., 2016*). For MalE, SBD1, SBD2 and OppA experiments were carried out in buffer A supplemented with 1 mM Trolox and 10 mM Cysteamine (Merck).

## Analysis of fluorescence trajectories

Time-traces were analysed by integrating the detected red and green photon streams in time-bins as stated throughout the text. Only traces lasting longer than 50 time-bins, having on average more than 10 photons per time-bin that showed clear bleaching steps, were used for further analysis. The number of analysed molecules, transitions and the total observation time are indicated in *Supplementary file 4*. The apparent FRET per time-bin was calculated by dividing the red photons by the total number of photons per time-bin. The state-trajectory of the FRET time-trace was modelled by a Hidden Markov Model (HMM) (*Rabiner and Lawrence, 1990*). For this an implementation of HMM was programmed in Matlab (MathWorks), based on the work of Rabiner (*Rabiner and Lawrence, 1990*). In the analysis, we assumed that the FRET time-trace (the observation sequence) can be considered as a HMM with two states having a one-dimensional Gaussian-output distribution. The Gaussian output-distribution of state $i$ ($i$=1, 2) is parameterized by its mean and variance. The parameters $\lambda$ (transition probabilities that connect the states and parameters of output-distribution), given the observation sequence, was found by maximizing the likelihood function. This was iteratively done using the Baum-Welch algorithm (*Baum and Petrie, 1966*). Care was taken to avoid floating point underflow and was done as described (*Rabiner and Lawrence, 1990*). With the inferred parameters $\lambda$, the most probable state-trajectory is then found using the Viterbi algorithm (*Viterbi, 1967*). The time spent in each state (open, closed) was inferred from the most probable state-trajectory, an histogram was made and the mean time spent in each state was calculated.

## Ensemble FRET

Fluorescence spectra of labeled SBD1 and SBD2 proteins were measured on a scanning spectrofluorometer (Jasco FP-8300; $\lambda_{ex}$ = 552 nm, 5 nm excitation and emission bandwidth; 3 s integration time). The apparent FRET efficiency was calculated via $I_{acceptor}/(I_{acceptor} + I_{donor})$, where $I_{acceptor}$ and $I_{donor}$ are fluorescence intensities around the emission maxima of the acceptor (660 nm) and donor fluorophore (600 nm), respectively. Measurements were performed at 20°C with ~200 nM labeled protein dissolved in buffer A.

## Acknowledgements

We thank H Jung, D Griffith and M Wiertsema for discussions and reading of the manuscript.

## Additional information

### Funding

| Funder | Grant reference number | Author |
|---|---|---|
| European Commission | 638536 | Thorben Cordes |
| Deutsche Forschungsgemeinschaft | GRK2062/1 (C03) | Thorben Cordes |
| Deutsche Forschungsgemeinschaft | SFB863 (A13) | Thorben Cordes |
| National Health and Medical Research Council | 1080784 | Christopher A McDevitt |
| National Health and Medical Research Council | 1122582 | Christopher A McDevitt |
| Nederlandse Organisatie voor Wetenschappelijk Onderzoek | 722.012.012 | Giorgos Gouridis |
| H2020 European Research Council | ERC Advanced 670578 | Bert Poolman |
| Australian Research Council | DP170102102 | Christopher A McDevitt |
| Australian Research Council | FT170100006 | Christopher A McDevitt |
| European Molecular Biology Organization | ALF 47-2012 | Giorgos Gouridis |

The funders had no role in study design, data collection and interpretation, or the decision to submit the work for publication.

### Author contributions

Marijn de Boer, Conceptualization, Data curation, Software, Formal analysis, Validation, Investigation, Visualization, Methodology, Writing—original draft, Writing—review and editing; Giorgos Gouridis, Conceptualization, Formal analysis, Funding acquisition, Validation, Investigation, Methodology, Writing—original draft, Writing—review and editing; Ruslan Vietrov, Formal analysis, Investigation, Methodology; Stephanie L Begg, Gea K Schuurman-Wolters, Formal analysis, Investigation; Florence Husada, Nikolaos Eleftheriadis, Investigation; Bert Poolman, Conceptualization, Resources, Supervision, Funding acquisition, Validation, Methodology, Writing—original draft, Writing—review and editing; Christopher A McDevitt, Conceptualization, Resources, Supervision, Funding acquisition, Validation, Investigation, Methodology, Writing—original draft, Writing—review and editing; Thorben Cordes, Conceptualization, Resources, Supervision, Funding acquisition, Validation, Methodology, Writing—original draft, Project administration, Writing—review and editing

### Author ORCIDs

Marijn de Boer http://orcid.org/0000-0002-0067-9020
Bert Poolman http://orcid.org/0000-0002-1455-531X
Christopher A McDevitt http://orcid.org/0000-0003-1596-4841
Thorben Cordes http://orcid.org/0000-0002-8598-5499

### Decision letter and Author response

Decision letter https://doi.org/10.7554/eLife.44652.036
Author response https://doi.org/10.7554/eLife.44652.037

## Additional files

### Supplementary files

• Supplementary file 1. P-values of two-way Kolmogorov-Smirnov test on the solution-based smFRET data.
DOI: https://doi.org/10.7554/eLife.44652.030

• Supplementary file 2. Primer sequences of all protein constructs used in this study.
DOI: https://doi.org/10.7554/eLife.44652.031

• Supplementary file 3. Apparent FRET efficiency values of solution-based measurements.
DOI: https://doi.org/10.7554/eLife.44652.032

• Supplementary file 4. Statistics of confocal scanning experiments of immobilized molecules.
DOI: https://doi.org/10.7554/eLife.44652.033

• Transparent reporting form
DOI: https://doi.org/10.7554/eLife.44652.034

### Data availability

Data generated or analysed during this study are included in the manuscript and supporting files. Source data files are available for smFRET histogrammes, representative smFRET time-traces and smFRET dwell-time histogrammes as shown in the manuscript. Primer sequences for created protein mutants are included.

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
