## [Decision Letter]

Thank you for submitting your article "Conformational and dynamical plasticity in substrate-binding proteins underlies selective transport in ABC importers" for consideration by *eLife*. Your article has been reviewed by 3 peer reviewers, including Baron Chanda as the Reviewing Editor and Reviewer #1, and the evaluation has been overseen by Richard Aldrich as the Senior Editor. The following individuals involved in review of your submission have agreed to reveal their identity: Marcel P. Goldschen-Ohm (Reviewer #3).

The reviewers have discussed the reviews with one another and the Reviewing Editor has drafted this decision to help you prepare a revised submission.

Summary:

This manuscript compares the interaction of both cognate and non-cognate ligands with the substrate bidding proteins (SBPs) for several different ABC importers. Although the smFRET technique employed is low resolution – monitoring only one or a few distances – it was able to detect distinct conformational states for SBPs bound to different ligands. Comparing the open/closed state of the SBP, lifetime in the bound state and transport assays led the authors to propose a model for how cognate (and non-cognate) ligands are recognized and transported (or not) through several different layers of recognition – SBP closing and ability to interact with the translocator, ligand release for transport, and some additional yet-unknown ligand-translocator interaction. The work is notable for the extensive comparisons that highlight commonalities and differences in ligand-SBP interaction, and it is this aspect that makes the work of broad interest.

Essential revisions:

1) Discussions of the different FRET states examine a difference between FRET states mean obtained from Gaussian fitting, for example in Figure 2. However, what is lacking is a discussion of whether these distributions are actually different, not just the mean values from fitting. Where any statistical tests performed to ask whether the distributions are indeed statistically different from one another and not a consequence of fitting (i.e. a two-way Kolmogorov Smirnov test)?

2) Given the variety of FRET states discovered across a many of proteins, it is surprising that there is no relation of these values into physical distances of the protein. As there are clearly known structures (i.e. Figure 1), have the authors considered computing these values to explore whether structural movements are indeed similar across the family? Overall, a discussion of how large/small these conformational changes are would be interesting.

3) The information laid out in Table 3 and Table 4 is a very helpful feature of the manuscript. However, it would be interesting to see a comparison of the FRET values obtained between both solution and tethered measurements (i.e MalE) to see how experimental set-ups may vary these values. Further, this would ensure that tethering a protein does not modify function in any way.

4) Discussion and interpretation of the results shown in Figure 6H. Based on earlier data in the manuscript, the authors hypothesize that faster ligand release will lead to faster transport. However, this figure shows exactly the opposite trend. The final paragraph of the Results section reads as if this figure is consistent with the hypothesis and does not make clear that the actual data contradicts this hypothesis. In the Discussion section and Figure 7, the authors invoke an additional ligand-translocator interaction to explain this result. Of course some sort of mechanism beyond what was directly observed with these experiments must be operating, but a clearer statement of the inconsistency of this result with the original hypothesis and an expanded discussion of the limitations of the current data set and multiple possible explanations (some of which are mentioned in the Discussion section but not included in Figure 7) would significantly clarify the strengths and limitations of the model in Figure 7.

5) They size hypothesis for non-transport of large ligands by MalFGK2. Are the sizes of the cavity in MalFGK2 vs MalE in the available crystal structures consistent with this hypothesis that the ligands are too big for the translocator, but are bound by MalE?

6) One issue regarding the observation of a few rare high-FRET events in the absence of ligand is that it is not clear to me that the authors can rule out that these events are simply rare binding events from non-chelated ligand. Maybe a titration of the chelator could get at this. In any event, a bit more discussion of the interpretation of these observations should be added.

---

## [Author Response]

Essential revisions:1) Discussions of the different FRET states examine a difference between FRET states mean obtained from Gaussian fitting, for example in Figure 2. However, what is lacking is a discussion of whether these distributions are actually different, not just the mean values from fitting. Where any statistical tests performed to ask whether the distributions are indeed statistically different from one another and not a consequence of fitting (i.e. a two-way Kolmogorov Smirnov test)?

Our strategy is based on the mathematical fact that when the true mean (i.e. the expectation value) of two distributions are different, then also the entire distributions are different. Statistical difference of the mean was judged by comparison of their 95% confidence interval. However, we agree that no non-parametric tests, such as the two-way Kolmogorov Smirnov test, were performed to judge whether differences exist between the FRET distributions. We are grateful for the suggestion and implemented a discussion on the significance of the differences in the distributions in the revised manuscript.

For all solution-based FRET experiments, in which conclusions were drawn on basis of the difference or similarity of the FRET states, we performed the two-way Kolmogorov Smirnov test as suggested by the reviewers for inclusion to the revised manuscript. *P*-values are provided in Supplementary file 1. We observe that when the fitted mean of two distributions are significantly different also the entire distributions are significantly different and vice versa. This implies that our conclusions concerning FRET states are not a consequence of fitting bias, as similar conclusions are drawn based on the non-parametric test of the entire distribution. The Result section was adapted to include this discussion.

2) Given the variety of FRET states discovered across a many of proteins, it is surprising that there is no relation of these values into physical distances of the protein. As there are clearly known structures (i.e. Figure 1), have the authors considered computing these values to explore whether structural movements are indeed similar across the family? Overall, a discussion of how large/small these conformational changes are would be interesting.

We agree that a discussion on the quantitative values of the conformational changes was lacking. We extended Table 3 to include the inter-dye distance changes of all SBPs and their ligands. Throughout the Result section we included discussions on the magnitude of the inter-dye distance changes. Noteworthy, crystal structures are only available for a few ligands/proteins and the changes determined via FRET agree in general well with the calculated Cα-Cα distance changes in the X-ray crystal structures (see Table 3).

3) The information laid out in Table 3 and Table 4 is a very helpful feature of the manuscript. However, it would be interesting to see a comparison of the FRET values obtained between both solution and tethered measurements (i.e MalE) to see how experimental set-ups may vary these values. Further, this would ensure that tethering a protein does not modify function in any way.

A detailed comparison of the MalE FRET states of surface-tethered and freely diffusing proteins is now included (new Figure 6—figure supplement 1). The average apparent FRET efficiencies obtained from solution and surface measurements are in good agreement. Please note that the width of the distributions cannot be compared due to the different nature of the data. The MalE conformational plasticity, as observed for the freely diffusing proteins with different maltodextrin substrates (Figure 2), was also observed when MalE was surface-tethered as shown in the new figures.

Ensuring that surface-tethering does not modify the biochemical activity of the protein (in our case the binding affinity for ligands) is an important point. To substantiate the claim that surface-tethering does not modify the proteins, we determined and compared the dissociation constant K_D_ of tethered- and freely diffusing labelled proteins (single-molecule data in new Figure 2—figure supplement 1; an overview of all K_D_ values are shown in Table 1). We performed this comparison for all proteins studied in our work, i.e. SBD1, SBD2, MalE, OppA and OpuAC, except for PsaA. The latter could not be studied since its low K_D_ poses the fundamental difficulty of dealing with small metal contaminations under single-molecule conditions. The necessity for addition of EDTA made it difficult to determine the actual metal concentration and accordingly it was not possible to derive the actual K_D_. For all other five proteins the K_D_ values of surface-tethered and freely diffusing proteins and labelled and unlabeled proteins agree well. Taken together, this provides strong indication that surface-tethering and labelling does not modify the ligand-binding process and the protein conformations.

4) Discussion and interpretation of the results shown in Figure 6H. Based on earlier data in the manuscript, the authors hypothesize that faster ligand release will lead to faster transport. However, this figure shows exactly the opposite trend. The final paragraph of the Results section reads as if this figure is consistent with the hypothesis and does not make clear that the actual data contradicts this hypothesis. In the Discussion section and Figure 7, the authors invoke an additional ligand-translocator interaction to explain this result. Of course some sort of mechanism beyond what was directly observed with these experiments must be operating, but a clearer statement of the inconsistency of this result with the original hypothesis and an expanded discussion of the limitations of the current data set and multiple possible explanations (some of which are mentioned in the Discussion section but not included in Figure 7) would significantly clarify the strengths and limitations of the model in Figure 7.

We thank the referee for bringing up these discussion points that were included in the revised manuscript (Results section and Discussion section of the revised manuscript).

In the case of Zn^2+^ transport, the conformational dynamics of PsaA was altered by the D280N mutation; a faster opening of PsaA(D280N) (compared to wildtype) result in a faster rate of transport. Similar observations were made for maltose transport via MalE and MalE(A96W/I329W). In both experiments the SBP lifetime versus transport rate of the *same* substrate is compared and a negative correlation is observed. From a mechanistic point of view these findings are logical. When the substrate-bound SBP has docked onto the transporter, it has to open to donate the substrate to the translocator. When only this openings step is faster, via the D280N or A96W/I329W mutation(s), the corresponding transport rate of the substrate will increase and vice versa.

In Figure 6H, the MalE lifetime versus transport activity of *different* substrates are compared and a positive correlation is observed. The different substrates alter the MalE openings rate (Figure 6A), so from our earlier results based on the examination of the same substrate, a faster opening would increase the donation of substrate to the translocator. If all other steps were independent of the specific substrate to be transported, a negative correlation would be expected to hold. However, a positive correlation between the transport rate and the SBP lifetime of different substrates provides evidence that the kinetics of certain other rate-determining steps are substrate-dependent. Faster transport can arise when certain substrates trigger these steps more efficient than others, thereby overcoming the slower opening of MalE.

In summary, when only the SBP conformational dynamics are altered, for example when transport of the *same* substrate is compared via SBPs with different conformational dynamics, a negative correlation between SBP lifetime and the rate of transport exists. When both the SBP conformational dynamics and other rate-determining steps are altered, for example by comparing transport of *different* substrates, a deviation from the negative correlation is well possible. We thus propose that the opposite behaviors reflect differences in the rate-determining steps in the overall transport cycle.

We agree with the reviewers that our data does not allow to judge which exact steps of the transport cycle are substrate-dependent. A more extended discussion on the possible steps is included in the revised manuscript (Results section and Discussion section of the revised manuscript) and altered the model of Figure 7 accordingly by replacing the phrase ‘enhanced-translocator interaction – faster transport’ with a more general statement ‘kinetics of downstream steps are substrate-dependent’. Furthermore, a clear statement of the opposite results (positive/negative correlation) and its mechanistic interpretation(s) is now included (Results section and Discussion section of the revised manuscript).

5) They size hypothesis for non-transport of large ligands by MalFGK2. Are the sizes of the cavity in MalFGK2 vs MalE in the available crystal structures consistent with this hypothesis that the ligands are too big for the translocator, but are bound by MalE?

In the crystal structures of MalE bound to maltose, maltotriose and maltotetraose it is shown that, in agreement with our FRET results (Figure 2—figure supplement 3A), the protein is in a closed conformation. The binding-pocket in the crystal structures can completely engulf 4 (i.e. maltotetraose) or possibly 5 (i.e. maltopentaose) glucosyl units (Quiocho et al., 1997), while longer maltodextrins would have to protrude from the binding pocket into the solution. Protruding from the fully closed MalE conformation seems still possible, however, based on our findings we conclude that large maltodextrins, such as β-cyclodextrin, maltoheptaose, maltooctaose and maltodecaose, induce only partial closing of MalE (Figure 2—figure supplement 3Aand Figure 4F). This structural flexibility of the protein would allow the binding pocket to adapt its size and allow the ligand to protrude more easily from the protein into the solution.

Based on the finite size of the translocation pathway of any ABC importer, it is expected that size restrictions of some sort exist for the transported substrates. Indeed, Oldham et al., (2013) showed by modelling a maltoheptaose molecule in the substrate cavity of the full transporter complex that probably only maltodextrins as large as maltoheptaose would fit, and longer maltodextrins, such as maltooctaose and maltodecaose, would experience steric hindrance. Moreover, and contrary to the binding pocket of MalE, in the crystal structures of the full transporter complex the ligand cannot easily protrude into the solution, as the MalE seals the binding cavity of the translocator (Oldham et al., 2007). This explains why MalE can accommodate large maltodextrins like maltodecaose and the translocator cannot. The manuscript was adapted to include this discussion (Discussion section of the revised manuscript).

6) One issue regarding the observation of a few rare high-FRET events in the absence of ligand is that it is not clear to me that the authors can rule out that these events are simply rare binding events from non-chelated ligand. Maybe a titration of the chelator could get at this. In any event, a bit more discussion of the interpretation of these observations should be added.

We thank the reviewer for this good suggestion. We agree that a chelator titration is required to make our argument that we examine true apo protein conformational dynamics and not rare binding events arising from non-chelated ligand.

We measured the protein conformational dynamics in the presence of 10-20 μM and in the presence of 4 to 10 times lower concentration of unlabeled protein. We conducted these new experiments for OppA, SBD1 and SBD2 since only for those proteins conformational dynamics were observed in the absence of ligand and in the presence of unlabeled protein. We found that the frequency and the lifetime of the high FRET transitions are not statistically significantly different when a lower concentration of unlabeled protein was used (new Figure 3—figure supplement 2). This provides strong evidence that the FRET transitions arise due to intrinsic conformational dynamics of the SBP, and we thus rule out that these transitions originate from rare binding events from any non-chelated ligand. Additional data was added (new Figure 3—figure supplement 2) and the Result section was adapted to include this discussion.